# SpeeCheck: Self-Contained Speech Integrity Verification via Embedded Acoustic Fingerprints

## Abstract

Advances in audio editing have made public speeches increasingly vulnerable to malicious tampering, raising concerns for social trust. Existing speech tampering detection methods remain insufficient: they often rely on external references or fail to balance sensitivity to attacks with robustness against benign operations like compression. To tackle these challenges, we propose SpeeCheck, the first learning-based self-contained speech integrity verification framework. SpeeCheck can (i) effectively detect tampering attacks, (ii) remain robust under benign operations, and (iii) enable direct verification without external references. Our approach begins with utilizing multiscale feature extraction to capture speech features across different temporal resolutions. Then, it employs contrastive learning to generate fingerprints that can detect modifications at varying granularities. These fingerprints are designed to be robust to benign operations, but exhibit significant changes when malicious tampering occurs. To enable self-contained verification, these fingerprints are embedded into the audio itself via a watermark. Finally, during verification, SpeeCheck retrieves the fingerprint from the audio and checks it with the embedded watermark to assess integrity. Extensive experiments demonstrate that SpeeCheck reliably detects tampering while maintaining robustness against common benign operations. Real-world evaluations further confirm its effectiveness in verifying speech integrity. The code and demo are available at https://speecheck.github.io/SpeeCheck/.

## 1 Introduction

Audio serves as an important information carrier that is widely used in news reporting, legal evidence, and public statements. However, the rapid development of audio editing tools (Wang et al., 2023) and text-to-speech (TTS) generation models (Wang et al., 2017; Ping et al., 2018; Huang et al., 2023; Du et al., 2024; Chen et al., 2024) has significantly lowered the technical barriers for speech manipulation and synthesis. While these techniques benefit content creation and entertainment, they also enable attackers to tamper speech content with ease. Public speeches and statements, especially made by influential figures, have become prime targets for attacks due to their huge social impact (Reuters, 2023; Post, 2024). Tampered speech can cause the spread of misinformation, undermine public trust, and even threaten social stability. Moreover, the prevalence of social media platforms accelerates the circulation of tampered audio, posing challenges to ordinary people in identifying authenticity from numerous sources. Currently, verifying the truth often requires cross-checking information across multiple social media platforms, a process that is time-consuming and prolongs the spread of misinformation. These challenges highlight a critical need: Is it possible to proactively protect publicly shared speech against tampering attacks while still allowing it to be freely stored, distributed, and reshared?

Existing approaches against speech tampering can be categorized into two groups: passive detection and proactive protection. Passive detection methods (Rodríguez et al., 2010; Yang et al., 2008; Pan et al., 2012; Blue et al., 2022; Leonzio et al., 2023) rely on deep binary classifiers trained to identify artifacts introduced by tampering. While they show reasonable performance against known attacks, their sensitivity to unseen or sophisticated manipulations remains limited. Moreover, passive detection alone cannot verify whether the speech content originates from the claimed speaker,

leaving systems vulnerable to impersonation-based attacks (Khan et al., 2022). Proactive protection methods verify integrity by extracting auxiliary information from the original audio and reusing it during verification. Early approaches compute cryptographic hashes (Steinebach & Dittmann, 2003; Zakariah et al., 2018) or embed fragile watermarks (Renza et al., 2018; Sripradha & Deepa, 2020; Zhang et al., 2024), which are very sensitive to any modification and therefore can reliably detect minor changes. However, cryptographic hashes require storing or transmitting external reference values, which prevents independent verification from the published audio. Fragile watermarking embeds a highly sensitive pattern into the signal and can enable self-contained verification, but the watermark is easily destroyed by benign operations, which restricts applicability in real-world distribution scenarios. To address this limitation, semi-fragile watermarking (Masmoudi et al., 2020; Wang et al., 2019; Wang & Fan, 2010) and watermark-fingerprint schemes (Gomez et al., 2002; Gulbis et al., 2006; Steinebach & Dittmann, 2003) embed carefully selected bits or perceptual hashes into transform domains (Zhang et al., 2021) so that they survive expected benign processing but are damaged by local tampering. This design, however, requires tight coupling between the embedding pattern and the assumed attack model (Yu et al., 2017), and adapting to new operations often implies redesigning the watermarking scheme or the handcrafted features. More recently, neural audio watermarking has been used for proactive defense, for example by embedding speaker embeddings to detect voice conversion (Ge et al., 2025), or by using watermark payloads to flag AI-content (Roman et al., 2024; Chen et al., 2023) or cloned speech (Liu et al., 2024a). These methods provide proactive protection against specific threats, but there is still no unified speech integrity verification solution that can easily handle diverse tampering attacks while remaining compatible with real-world distribution scenarios.

To address the issues above, a desired speech verification design should have the following properties: (1) **Convenient to use**: the integrity of the speech can be verified directly from the published audio without requiring external references. (2) **Sensitive to tampering attacks**: it can reliably detect any malicious edits, including subtle semantic (e.g., can ⇔ cannot) or speaker-related (e.g., timbre) changes. (3) **Robust to benign operations**: it remains stable under typical benign audio operations, especially commercial-off-the-shelf codecs (e.g., AAC in Instagram/TikTok), ensuring usability in sharing and distribution. Therefore, in this paper, we propose SpeeCheck, a proactive acoustic fingerprint-based speech verification framework that jointly exploits semantic content and speaker identity. Instead of controlling robustness and fragility through the watermark embedding scheme or handcrafted features, we adopt a decoupled architecture. A robust neural watermark is used purely as a carrier, while integrity verification is governed entirely by the embedded fingerprint. Specifically, the fingerprint is generated by a multiscale feature extractor that captures speech characteristics across different temporal resolutions. By using contrastive learning, the fingerprint is designed to be stable under benign operations, yet to change significantly when malicious tampering occurs. To enable self-contained verification, the generated fingerprints are embedded into the speech signal by segment-wise watermarking. Without access to the original authentic speech, SpeeCheck can recover the fingerprint from the published audio and check it with the fingerprint embedded in the watermark payload to verify the integrity. Our main contributions are summarized as follows.

- We present SpeeCheck, a learning-based self-contained speech integrity verification framework with a decoupled fingerprint–watermark architecture that is easy to extend and adapt to new operations.

- We develop a discriminative fingerprint generator that extracts multiscale features and applies contrastive learning to produce binary fingerprints, which are robust to benign operations yet sensitive to malicious manipulations.

- We evaluate SpeeCheck through extensive experiments on public speech datasets and a real-world dataset constructed for this study. The results demonstrate high effectiveness in detecting diverse tampering attacks while maintaining robustness against benign operations in practical scenarios.

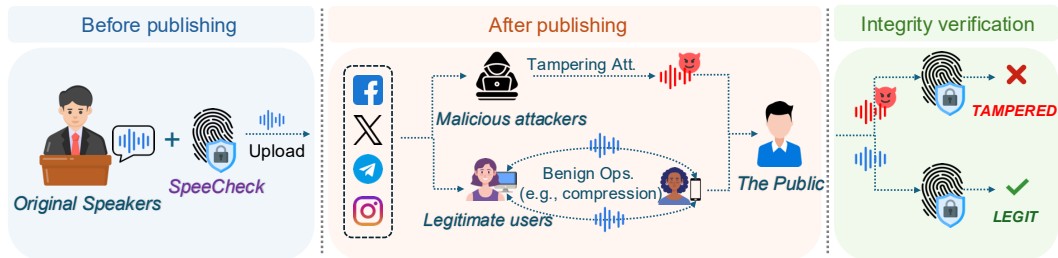

Figure 1: System overview of the proposed SpeeCheck.

## 2 MOTIVATION

### 2.1 PROBLEM DEFINITION

As shown in Figure 1, the scenario considered in our study includes four parties: 1) **Original speakers**, such as public institutions and celebrities, who publish statements or speeches on social media platforms. 2) **Legitimate users**, who help disseminate these audio recordings by downloading or reposting them. 3) **Malicious attackers**, which employ audio editing or voice conversion techniques to alter either the semantic content or the speaker's identity. 4) **The public**, who are exposed to conflicting audio sources, requires a reliable method to verify the integrity of a given speech recording.

### 2.2 MALICIOUS AND BENIGN AUDIO OPERATIONS

We define malicious audio tampering as intentional audio modifications that alter the semantic content or speaker identity. Typical malicious operations include audio splicing, deletion, substitution, silencing, text-to-speech (TTS) synthesis, and voice conversion. In contrast, benign operations refer to common audio transformations that occur during legitimate processes such as storage, transmission, or distribution. Examples include compression, reencoding, resampling, and noise suppression, none of which impact the semantic content or speaker identity. A detailed distinction between malicious and benign audio operations, along with specific examples, is provided in Appendix B.2.

### 2.3 LIMITATIONS OF ACOUSTIC FEATURE SIMILARITY

An intuitive approach for speech verification is to compare the acoustic similarity between the published audio and its original version.

Following this intuition, we analyzed similarity scores between the original audio and three types of modifications: benignly processed variants ("Benign"), maliciously modified variants ("Malicious"), and unrelated audio samples ("Cross"). Figure 2 presents cosine similarity distributions computed using wav2vec embeddings (Baevski et al., 2020). The significant overlap between benign and malicious similarity distributions demonstrates that acoustic feature similarity alone is insufficient to determine the types of modification operations. Similar results are observed using traditional acoustic feature Mel-frequency cepstral coefficients (MFCC) (Davis & Mermelstein, 1980), detailed in Appendix C.3. This limitation arises because malicious

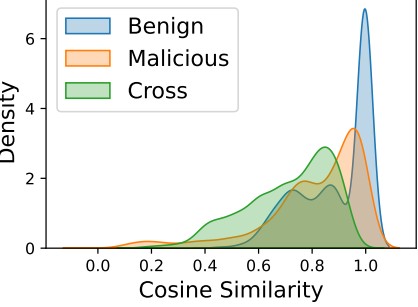

Figure 2: Probability distributions of the wav2vec embedding similarity to the original audio under different modifications.

operations, even significantly altering the content, may introduce minimal acoustic changes. For instance, modifying the phrase "do not" to "do" in a 20-second speech affects only 0.2 seconds, and similarity remains more than 99%, while causing substantial semantic alteration. Moreover, this method requires access to the authentic audio, which is impractical in real-world scenarios. These limitations highlight two key challenges:

**Challenge 1: Insufficient sensitivity to semantic tampering attacks.** Acoustic feature-based similarity methods fail to distinguish benign operations from malicious ones, because they are not sensitive enough to semantic tampering attacks.

**Challenge 2: Dependence on the original authentic audio.** Acoustic similarity assessments require the original authentic audio as a reference, which is not always applicable in practice.

## 2.4 LIMITATIONS OF HASH-BASED INTEGRITY VERIFICATION

Another common class of integrity verification methods is hash-based verification. Cryptographic hashing (Menezes et al., 2018) (e.g., SHA256, MD5) is widely used in practice for digital file integrity. Given an audio file, these functions produce a short digest that changes completely even when a single bit is modified. This extreme sensitivity is ideal for strict file integrity, but it is too strict for speech content integrity: benign operations such as compression or resampling already produce a completely different digest, even though they do not alter the semantic content or the speaker identity.

To relax this sensitivity while retaining content awareness, perceptual hashing (Zhang et al., 2021; Li et al., 2021) has been proposed. These methods extract handcrafted content descriptors (such as cepstral coefficients (Zhang et al., 2021), spectral envelopes (Zhang et al., 2018), or time–frequency energy patterns) and convert them into compact binary hashes. Their robustness to benign processing and their sensitivity to malicious tampering are determined by design choices such as which features are used and how they are quantized. As a result, a given perceptual hash is typically tailored to a specific set of operations, and adapting it to new codecs, platforms, or tampering attacks often requires redesigning the feature extractor or the quantization rule. In addition, hash-based verification requires externally stored or transmitted reference hash values for verification, which introduces practical complexity in real-world speech sharing and forwarding. These limitations lead to the following challenges:

**Challenge 3: Difficulty in balancing robustness and sensitivity.** Cryptographic hashes are overly sensitive to any modification, while perceptual hashes rely on handcrafted features whose robustness to benign operations and sensitivity to tampering must be manually tuned for specific operation sets, and are hard to adapt to new operations.

**Challenge 4: Dependence on external reference hash values.** Hash-based verification depends on externally stored or transmitted hash values, which prevents self-contained verification from a single audio file and introduces extra overhead and inconvenience for online speech distribution.

## 3 METHODOLOGY

### 3.1 SPEECHECK OVERVIEW

To address these challenges, we propose SpeeCheck, a proactive speech integrity verification design, which is (i) sensitive to tampering attacks, (ii) robust to benign operations, and (iii) convenient to use by the public since it verifies the published speech audio's integrity in a self-contained manner. As the sketch shown in Figure 3, SpeeCheck consists of two stages: fingerprint generation and dual-path integrity verification.

The speech fingerprint generation in SpeeCheck has five steps: (1) Frame-Level Feature Encoding (Speech to Representation): raw speech is encoded into frame-level representations that preserve acoustic information; (2) Multiscale Acoustic Feature Extraction (Representation to Vector): the frame-level representations are first processed into contextual features, then aggregated at multiple temporal resolutions, and finally attentively pooled into a fixed-dimensional vector that summarizes the entire utterance; (3) Contrastive Fingerprint Training (Vector to Fingerprint): the vector is optimized to be robust to benign operations, and sensitive to tampering attacks using contrastive learning; (4) Binary Fingerprint Encoding (Fingerprint to Bit): the trained fingerprint is discretized into a binary representation; (5) Segment-Wise Watermarking (Bit to Watermark): the binary fingerprint is embedded into the original audio through segment-wise watermarking, making the fingerprint self-contained.

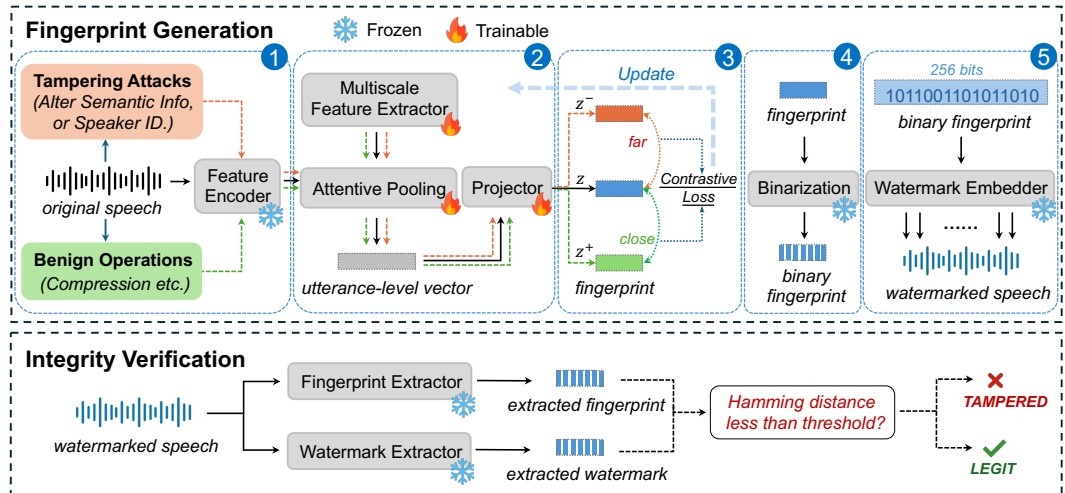

Figure 3: A sketch of the proposed SpeeCheck design including speech fingerprint generation (top) and integrity verification (bottom).

The integrity verification in SpeeCheck independently performs two parallel paths on the published audio: (1) regenerating the fingerprint via the same extraction pipeline, and (2) extracting the embedded watermark via the watermark decoder. The two resulting binary codes are then compared using Hamming distance to determine whether the speech has been attacked.

## 3.2 FINGERPRINT GENERATION AND WATERMARKING

**Step 1. Frame-Level Feature Encoding (Speech to Representation)** We utilize the pre-trained wav2vec 2.0 model (Baevski et al., 2020) to extract frame-level representations from the original audio before publishing. This step serves as a necessary preprocessing stage for fingerprint generation. It converts continuous waveform signals into structured sequences of frame-level representations that preserve essential acoustic information. These representations have demonstrated effectiveness in downstream tasks such as automatic speech recognition (Baevski et al., 2021) and speaker verification (Fan et al., 2021). Formally, the feature encoder $\varepsilon : \mathcal{X} \to \mathcal{Z}$ maps raw audio waveforms $\mathcal{X}$ to a sequence of latent representations $\mathbf{z}_1, \mathbf{z}_2, \ldots, \mathbf{z}_T$, where each $\mathbf{z}_t \in \mathbb{R}^{d_z}$ denotes the frame-level acoustic feature at time $t$, and $T$ is the total number of output frames.

**Step 2. Multiscale Acoustic Feature Extraction (Representation to Vector).** Given the frame-level representations $\mathbf{z}_1, \mathbf{z}_2, \ldots, \mathbf{z}_T$ obtained from Step 1, this step constructs a fixed-dimensional vector that summarizes the speech across different temporal granularities. The multiscale feature extractor $\mathcal{F}$ consists of two components: (a) a bidirectional long short-term memory (BiLSTM) network that transforms the input frame-level representations into contextual hidden states, and (b) a multiscale pooling operation that averages the hidden states over phoneme-, word-, and phrase-level windows (size 20, 50, and 100, respectively) , producing a sequence of multiscale features $\mathbf{h}_1, \mathbf{h}_2, \ldots, \mathbf{h}_K$ (see Appendix B.3 for examples).

To summarize these features into an utterance-level vector, we apply self-attentive pooling (Lin et al., 2017). This mechanism assigns higher weights to more informative components, with attention weight computed as: $w_n = \frac{\exp(\phi(h_n))}{\sum_{t=1}^{K} \exp(\phi(h_t))}$, where $\phi(\cdot)$ is a feedforward network. The weighted sum yields a fixed-dimension vector: $\mathbf{v}' = \sum_{n=1}^{K} w_n \cdot h_n$, which is referred to as the utterance-level vector. To obtain a more compact representation for fingerprint optimization, a projection module is applied to reduce the dimensionality of $\mathbf{v}'$, yielding the final fingerprint vector $\mathbf{v} \in \mathbb{R}^{d_v}$.

**Step 3. Contrastive Fingerprint Training (Vector to Fingerprint).** Given the fixed-length vector $\mathbf{v}$ obtained from Step 2, we optimize it to serve as a distinctive audio fingerprint that is robust to benign operations and sensitive to malicious tampering attacks. To this end, we adopt contrastive learning (Oord et al., 2018) to guide the training of all preceding modules. During the training, a batch of original speech samples is randomly selected, where each sample serves as an anchor. For

each anchor, we generate: positive pairs, consisting of the anchor and its benign variants (e.g., compression), and negative pairs, consisting of the anchor and its tampered variants (e.g., substitution). Detailed operations are listed in Appendix B.2. The contrastive loss is defined as

$$\mathcal{L}_c = -\frac{1}{B}\sum_{i=1}^{B}\frac{1}{P}\sum_{j=1}^{P}\log\frac{\exp\left(\tilde{\mathbf{v}}_i^{\text{Orig.}\top}\tilde{\mathbf{v}}_{i,j}^{\text{Benign}}/\tau\right)}{\sum\limits_{k=1,\ k\neq i}^{N}\exp\left(\tilde{\mathbf{v}}_i^{\text{Orig.}\top}\tilde{\mathbf{v}}_{i,k}/\tau\right)},\tag{1}$$

where $B$ is the number of anchors in the batch, $P$ is the number of benign variants per anchor, and $N$ denotes the total number of comparison samples for each anchor, including its own benign and tampered variants as well as embeddings from other anchors in the batch. $\tau$ is the temperature parameter. $\tilde{\mathbf{v}}_i^{\text{Orig.}}$ denotes the L2-normalized embedding of the $i$-th anchor, $\tilde{\mathbf{v}}_{i,j}^{\text{Benign}}$ denotes the embedding of its $j$-th benign variant, and $\tilde{\mathbf{v}}_{i,k}$ enumerates all embeddings in the batch, including benign, tampered and unrelated samples.

This contrastive learning above encourages the model to bring the anchor closer to its benign variants while pushing it away from tampered and unrelated samples in the embedding space. As a result, the fixed-length vector is optimized to serve as a distinctive audio fingerprint that is robust to benign operations while remaining sensitive to malicious tampering attacks.

**Step 4. Binary Fingerprint Encoding (Fingerprint to Bit).** To enable self-contained verification, we embed the generated fingerprint into the audio signal as a watermark. Since watermarking schemes typically support binary payloads and inevitably incur information loss, we design the fingerprint representation to preserve its discriminative power even after binarization. Specifically, we convert the continuous fingerprint vector $\mathbf{v} \in \mathbb{R}^{d_v}$ into a binary code $\mathbf{b} \in \{-1, +1\}^d$, which is more suitable for compact storage and fast retrieval via bit-wise comparison. To encourage the output to approach the bipolar extremes of -1 and +1 and thus reduce quantization error, we apply a `tanh` activation at the final projection layer. This is followed by a `sign` function to obtain the final binarized output. As demonstrated in Section 4.2, the binarized fingerprint retains its discriminative characteristics of $\mathbf{v}$, i.e., robust to benign operations while sensitive to tampering attacks. In adversarial settings where an attacker may manipulate the content, compute a new fingerprint, and attempt to re-embed it into the audio stream (replay-style attacks), this binarization step can be further secured by introducing a secret key, as detailed in Appendix D.2.

**Step 5. Segment-Wise Watermarking (Bit to Watermark).** To enable self-contained verification, the binary fingerprint must be embedded directly into the speech signal. We adapt the robust watermarking method AudioSeal (Roman et al., 2024) for this purpose. However, a key challenge arises from our high-capacity requirement. While AudioSeal is designed for short watermarks (i.e., 16 bits) for copyright protection, our task requires embedding much longer fingerprints (e.g., 256 bits). To meet this requirement, we extend the original AudioSeal with a segment-wise embedding strategy. Given an input waveform $\mathcal{X}$ of duration $T$ seconds and its binary fingerprint $\mathbf{b}$, both are divided into $N$ non-overlapping segments:

$$\mathcal{X} = [\mathcal{X}^{(1)}, \ldots, \mathcal{X}^{(N)}] \quad \text{and} \quad \mathbf{b} = [\mathbf{b}^{(1)}, \ldots, \mathbf{b}^{(N)}],\tag{2}$$

where each $\mathcal{X}^{(n)}$ spans $T/N$ seconds and each $\mathbf{b}^{(n)}$ contains $d/N$ bits. For each audio segment $\mathcal{X}^{(n)}$, we embed $\mathbf{b}^{(n)}$ into the Encodec embedding space and generate a watermark signal $\delta^{(n)}$. The watermarked segment is then formed as: $\tilde{\mathcal{X}}^{(n)} = \mathcal{X}^{(n)} + \delta^{(n)}$. Finally, the watermarked segments $[\tilde{\mathcal{X}}^{(1)}, \ldots, \tilde{\mathcal{X}}^{(N)}]$ are concatenated, yielding the final self-verifiable audio.

Notably, the watermark incurs only subtle perturbations. Our experiments in Appendix C.12 confirm that the acoustic fingerprint generated from the watermarked audio $\tilde{\mathcal{X}}$ remains consistent with the original fingerprint, while the embedded bits can still be reliably extracted without degradation.

### 3.3 Dual-Path Speech Integrity Verification

SpeeCheck employs a dual-path mechanism to assess the integrity of the published speech $\tilde{\mathcal{X}}$:

**Path A: Fingerprint Generation from Published Speech.** The published speech audio is processed using the same fingerprint generation pipeline described before. The fingerprint $\mathbf{b}'$ is computed as $\mathbf{b}' = \text{sign}(\mathcal{F}(\varepsilon(\tilde{\mathcal{X}})))$, where $\varepsilon$ and $\mathcal{F}$ denote the feature encoder and multiscale extractor, respectively, and $\text{sign}(\cdot)$ denotes the final binarization function.

**Path B: Watermark Extraction.** The published speech audio $\tilde{\mathcal{X}}$ is processed in the inverse manner of Step 5 to decode the embedded watermark (i.e., the original binary fingerprint). From each segment $\tilde{\mathcal{X}}^{(n)}$, we extract the bit chunk $\hat{\mathbf{b}}^{(n)}$ using the watermark decoder, and then reconstruct the full watermark as $\hat{\mathbf{b}} = [\hat{\mathbf{b}}^{(1)}, \hat{\mathbf{b}}^{(2)}, \ldots, \hat{\mathbf{b}}^{(N)}]$.

Finally, the integrity of the published audio is verified by comparing the generated fingerprint $\mathbf{b}'$ with the extracted watermark $\hat{\mathbf{b}}$. This is done by computing the Hamming distance as follows.

$$d_H(\mathbf{b}', \hat{\mathbf{b}}) \leq \theta \quad \Rightarrow \quad \text{Accept;} \quad \text{otherwise} \quad \text{Reject,}$$

where $\theta$ is a decision threshold set based on the validation set from public datasets.

## 4 EXPERIMENTS

### 4.1 EXPERIMENT SETUP

**Dataset.** To train and evaluate the performance of SpeeCheck, we use VoxCeleb1 (Nagrani et al., 2017), which includes over 150,000 utterances from 1,251 celebrities. These audio samples are collected from interviews and public videos, providing conditions that reflect real-world speech recordings. We further employ the test subset from LibriSpeech (Panayotov et al., 2015) dataset to assess the model generalization. Furthermore, to validate SpeeCheck's effectiveness under authentic scenarios, we build a real-world speech dataset and evaluate it after fine-grained editing and distribution across major social media platforms. More details about the datasets and the preprocessing steps are provided in Appendix C.2.

**Implementation details.** (i) Fingerprint model: We use Wav2Vec2.0 Base model[1] as the acoustic feature extractor. A two-layer BiLSTM with a hidden size of 512 follows the feature extractor. Multiscale pooling is used with window sizes of 20, 50, and 100 frames with a stride of 10 frames. A two-layer projection head then maps features into a 256-dimensional vector. (ii) Watermark model: AudioSeal model[2] is used to embed and extract fingerprints as watermark payloads. To improve the watermarking capacity, we divide both the carrier audio and the fingerprint into 16 segments. Each segment carries a 16-bit watermark, leading to a total payload of 256 bits per audio sample. (iii) Training: We exploit benign and malicious operations (see Appendix B.2) and the original audio samples for contrastive learning, with temperature set as 0.05. A cosine annealing learning rate schedule is used, gradually decreasing the learning rate from $1 \times 10^{-3}$ to $1 \times 10^{-5}$ over the training.

**Evaluation Metrics.** We evaluate SpeeCheck as a binary classification task, where benign operations are treated as the positive class and malicious ones as the negative class. To characterize the detector over different operating points, we sweep the decision threshold $\theta$ across the full range of Hamming distances and compute the receiver operating characteristic (ROC) curve. From this curve, we derive the area under the curve (AUC) and the equal error rate (EER). Afterwards, we select a single decision threshold $\theta^\star = 42$ on the validation set to balance robustness to benign operations and sensitivity to tampering. All reported true positive rate (TPR), false positive rate (FPR), true negative rate (TNR), and false negative rate (FNR) in Tables 1 and 2 are computed at this fixed $\theta^\star$. Formal definitions of these metrics are given in Appendix C.2.

### 4.2 RESULTS

**Robustness to benign operations.** Table 1 presents the performance of the proposed SpeeCheck in accepting published speech samples subjected to benign audio operations, as defined in Appendix B.2. We focus on evaluating how well SpeeCheck accepts positive samples with harmless modifications (TPR) and whether it mistakenly accepts maliciously tampered speech (FPR). For each benign operation listed in Table 1, we construct a balanced test set that contains all benign samples produced by this operation and an equal number of maliciously tampered samples, randomly drawn from the pool of attacks in Appendix B.2. On the test subsets of VoxCeleb1, SpeeCheck achieves an overall TPR of 99.15% and an FPR of 0.55%, demonstrating strong robustness to non-malicious transformations. For cross-dataset evaluation on LibriSpeech, using a model

---

[1]https://github.com/facebookresearch/fairseq/tree/main/examples/wav2vec
[2]https://github.com/facebookresearch/audioseal

Table 1: Results of benign operation (positive) acceptance on VoxCeleb and LibriSpeech.

| Operation | VoxCeleb | | | | LibriSpeech | | | | Semantic | Identity |
|---|---|---|---|---|---|---|---|---|---|---|
| | TPR | FPR | AUC | EER | TPR | FPR | AUC | EER | | |
| Compression | 99.80 | 1.01 | 99.77 | 1.11 | 96.98 | 1.21 | 99.51 | 2.82 | ✓ | ✓ |
| Reencoding | 99.80 | 0.20 | 100.00 | 0.20 | 99.40 | 2.41 | 99.89 | 1.51 | ✓ | ✓ |
| Resampling | 97.18 | 0.60 | 99.56 | 1.21 | 97.18 | 1.81 | 99.17 | 2.21 | ✓ | ✓ |
| Noise suppression | 99.80 | 0.40 | 99.99 | 0.20 | 99.40 | 2.21 | 99.84 | 1.41 | ✓ | ✓ |
| **Overall** | 99.15 | 0.55 | 99.83 | 0.68 | 98.24 | 1.91 | 99.60 | 1.99 | - | - |

Table 2: Results of malicious operation (negative) rejection on VoxCeleb and LibriSpeech.

| Operation | VoxCeleb | | | | LibriSpeech | | | | Semantic | Identity |
|---|---|---|---|---|---|---|---|---|---|---|
| | TNR | FNR | AUC | EER | TNR | FNR | AUC | EER | | |
| Deletion | 100.00 | 1.21 | 100.00 | 0.00 | 100.00 | 1.54 | 99.97 | 0.07 | ✗ | ✓ |
| Splicing | 100.00 | 0.67 | 100.00 | 0.00 | 100.00 | 1.74 | 99.99 | 0.17 | ✗ | ✓ |
| Silencing | 98.59 | 0.74 | 99.64 | 1.24 | 98.12 | 1.81 | 99.71 | 1.88 | ✗ | ✓ |
| Substitution | 97.79 | 0.87 | 99.81 | 1.11 | 93.03 | 1.81 | 99.03 | 3.72 | ✗ | ✓ |
| Reordering | 97.59 | 0.60 | 98.62 | 2.11 | 98.39 | 1.41 | 99.21 | 1.71 | ✗ | ✓ |
| Text-to-speech | 100.00 | 0.00 | 100.00 | 0.00 | 100.00 | 0.00 | 100.00 | 0.00 | ✗ | ✓ |
| Voice conversion | 99.40 | 0.00 | 100.00 | 0.00 | 97.80 | 0.00 | 100.00 | 0.00 | ✓ | ✗ |
| **Overall** | 99.08 | 0.74 | 99.80 | 0.61 | 97.98 | 1.48 | 99.69 | 1.28 | - | - |

trained on VoxCeleb1, the TPR/FPR slightly change to 98.24% and 1.91%, respectively, indicating good generalizability across datasets.

**Sensitivity to tampering attacks.** Table 2 evaluates the ability of SpeeCheck to reject malicious tampering attacks that alter the semantic content or speaker identity. For each tampering category listed in Table 2, we similarly form a balanced binary task that contains all samples generated by this attack and an equal number of benign samples. The considered attacks include simple audio editing (e.g., deletion) as well as advanced learning-based manipulations such as text-to-speech (TTS) synthesis and voice conversion, as detailed in Appendix B.2. In this setting, tampering operations (actual negatives) are expected to be rejected with a high true negative rate (TNR), while minimizing the false negative rate (FNR), which reflects incorrect rejection of benign samples. Notably, on the VoxCeleb (in-domain) dataset, SpeeCheck achieves an overall TNR of 99.08% and an FNR of 0.74%. On the LibriSpeech dataset, the system maintains strong performance with an overall TNR of 97.98% and an FNR of 1.48%. These results highlight SpeeCheck's strong sensitivity to tampering attacks. A more detailed breakdown by tampering strength (e.g., minor, moderate, and severe) is provided in Appendix C.5, and Appendix C.11 further analyzes the remaining false positives and false negatives.

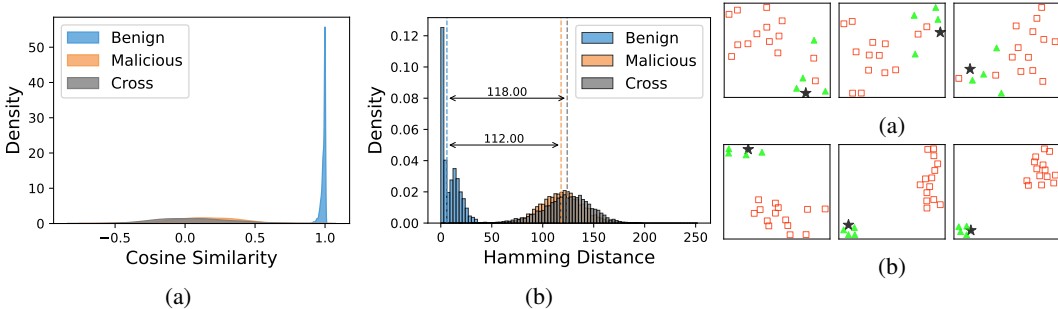

Figure 4: (a) Extracted feature similarity; (b) binarized fingerprint Hamming distance.

Figure 5: t-SNE visualizations of speech samples: (a) before training; (b) after training.

**Multiscale feature and binarized fingerprints analysis.** Figure 4 shows two analyses: (a) cosine similarity between extracted multiscale features and (b) Hamming distance between binarized

fingerprints. In Figure 4a, benign-original pairs yield high similarity values close to 1.0, while malicious-original and cross-original pairs are much lower, indicating that the learned multiscale features effectively capture the differences between benign operations and malicious tampering attacks. Here, "cross" refers to the arbitrarily selected unrelated audio samples. In Figure 4b, binarized fingerprints of benign processed samples yield low Hamming distances from their retrieved watermarks, whereas malicious and cross pairs have much larger distances, with a clear margin of about 112-118 bits. This indicates that the binarization process preserves discriminability and enables reliable separation of tampering and benign operations using a simple threshold.

Figure 5 shows the t-SNE visualizations of the extracted multiscale features before and after training. Specifically, Figure 5a and Figure 5b show the distribution of anchors (original speech), positives (after benign operations), and negatives (after malicious operations) in the latent space. Before training, anchor and positive samples are scattered and overlap with negatives, indicating poor separability. After training, anchors and positives form tight clusters, while negatives are clearly separated. This suggests that contrastive learning enables the multiscale feature extractor to learn embeddings that distinguish benign operations from malicious tampering, which explains the strong performance of SpeeCheck. Additional visualization evidences are provided in Appendix C.15.

Table 3: Detection accuracy on unseen benign operations and tampering attacks.

| Unseen Operation Type | Accuracy (%) | $d_{wm}$ | $d_{fp}$ |
|---|---|---|---|
| **Benign** | | | |
| Loudness Normalization | 100.00 | 0.96 | 0.08 |
| Room Reverb | 100.00 | 3.12 | 26.60 |
| Combined Benign | 100.00 | 4.64 | 8.29 |
| **Tampering** | | | |
| Voice Changer (Female) | 100.00 | 122.46 | 71.62 |
| Voice Changer (Male) | 100.00 | 125.38 | 67.85 |
| Combined Malicious | 100.00 | 119.32 | 75.08 |

**Out-of-Distribution and real-world generalization.** To evaluate SpeeCheck's generalization to unseen audio operations, we conduct out-of-distribution (OOD) experiments using operations not included in the training set. Table 3 reports detection accuracy at a fixed threshold ($\theta^\star = 42$), along with the Hamming distances of watermarks ($d_{wm}$) and fingerprints ($d_{fp}$) between modified and original audio. SpeeCheck achieves 100% accuracy on unseen benign transformations, including loudness normalization, room reverberation, and combined benign manipulations. Concurrently, it successfully detects sophisticated unseen attacks, including commercial voice changer tools (ElevenLabs, 2024) that alter speaker identity and combined malicious edits. These results highlight the effectiveness of SpeeCheck in identifying unseen audio operations. Specifically, for real-world recordings, we observe similar strong performance, where SpeeCheck reliably identifies all fine-grained tamperings. Detailed evaluation is shown in Appendix C.6.

Table 4: Performance of Deepfake detection at varying substitution ratios.

| Deepfake Ratio | Nes2Net | | | | SSL-AntiSpoofing | | | | SpeeCheck (ours) | | | |
|---|---|---|---|---|---|---|---|---|---|---|---|---|
| | TPR | FPR | AUC | EER | TPR | FPR | AUC | EER | TPR | FPR | AUC | EER |
| Substitute 10% | 54.27 | 30.08 | 65.60 | 39.84 | 57.11 | 19.52 | 72.58 | 33.00 | **82.97** | **0.40** | **99.57** | **3.21** |
| Substitute 25% | 88.01 | 20.12 | 90.54 | 16.46 | 71.54 | 19.72 | 81.10 | 25.96 | **98.20** | **0.40** | **99.95** | **0.40** |
| Substitute 50% | 98.78 | 2.03 | 99.89 | 1.83 | 79.27 | 12.27 | 89.29 | 17.71 | **100.00** | **0.40** | **100.00** | **0.30** |
| Substitute 75% | 100.00 | 0.00 | 100.00 | 0.00 | 93.50 | 10.26 | 97.25 | 9.05 | **100.00** | **0.40** | **100.00** | **0.00** |
| Substitute 90% | 100.00 | 0.00 | 100.00 | 0.00 | 95.12 | 3.82 | 98.97 | 4.63 | **100.00** | **0.40** | **100.00** | **0.00** |

**Deepfake detection comparison.** We further evaluate SpeeCheck as a deepfake detector[3]. Specifically, we compare SpeeCheck with two strong deepfake detectors, Nes2Net (Liu et al., 2025) and SSL-AntiSpoofing (Tak et al., 2022), which are built upon self-supervised speech representations and achieve competitive performance on recent ASVspoof challenges (Yamagishi et al., 2021). We use the zero-shot TTS model YourTTS (Casanova et al., 2022) to synthesize deepfake speech segments, and substitute them for varying proportions (10%, 25%, 50%, 75% and 90%) of the original

---

[3]To avoid confusion, SpeeCheck is used here for deepfake detection, where "positive" now refers to deepfake samples to be identified.

speech. The resulting deepfake utterances are then mixed with an equal number of clean utterances to ensure fair evaluation. From Table 4, all three methods become stronger when the substitution ratio increases. When 75–90% of an utterance is replaced by deepfake content, Nes2Net and SSL-AntiSpoofing already show very strong performance (for example, at 90% substitution Nes2Net reaches 100.00% TPR and 100.00% AUC, while SSL-AntiSpoofing obtains 95.12% TPR and 98.97% AUC with 4.63% EER). However, their performance degrades sharply when the substitution ratio becomes small. At 10% substitution, Nes2Net only achieves 65.60% AUC with 39.84% EER, and SSL-AntiSpoofing reaches 72.58% AUC with 33.00% EER, indicating limited sensitivity to subtle spoofing. In contrast, SpeeCheck maintains strong detection performance across all substitution levels. For moderate to high substitution ratios ($\geq$25%), it achieves near-perfect detection. Even in the most challenging case with only 10% deepfake substitution, SpeeCheck still reaches 82.97% TPR while keeping FPR as low as 0.40%, with 99.57% AUC and 3.21% EER. This trend is consistent with the "Minor Substitution" results reported in Table 10 (Appendix C.5). Since synthetic deepfake audio does not carry embedded watermarks, the fingerprint–watermark verification process becomes essentially random, which makes tampering easier to detect. Even minor substitutions alter the extracted fingerprint and disrupt the embedded watermark at the same time, leading to a mismatch and enabling reliable detection of tampering. Further analysis of each module's contributions is provided in the ablation studies in Appendix C.11.

Table 5: Comparison with audio fingerprinting methods.

| Method | VoxCeleb | | | | LibriSpeech | | | |
|---|---|---|---|---|---|---|---|---|
| | TPR | FPR | AUC | EER | TPR | FPR | AUC | EER |
| Chromaprint (Lalinský, 2010) | 93.06 | 16.30 | 92.34 | 12.58 | 81.09 | 13.43 | 85.38 | 16.68 |
| Renza et al. (2019) | 82.40 | 3.35 | 93.85 | 12.37 | 74.90 | 2.05 | 89.67 | 18.53 |
| Shi et al. (2020) | 60.20 | 8.30 | 79.88 | 26.12 | 59.75 | 8.55 | 80.93 | 26.95 |
| Zhang et al. (2018) | 85.65 | 17.40 | 93.07 | 15.85 | 79.38 | 12.73 | 91.96 | 16.70 |
| Zhang et al. (2021) | 92.05 | 6.14 | 97.43 | 7.24 | 90.59 | 8.65 | 96.94 | 9.05 |
| **SpeeCheck (only fingerprint)** | 99.32 | 0.52 | 99.98 | 0.55 | 98.24 | 2.11 | 99.67 | 1.96 |

**Audio fingerprinting comparison.** We benchmark the fingerprint generated by SpeeCheck for integrity verification. Since existing audio fingerprinting and perceptual hashing schemes for speech authentication are non-learning-based and rely on handcrafted features, we select five representative methods and evaluate them under the same integrity verification protocol. Chromaprint (Lalinský, 2010) is a widely used open-source audio fingerprinting system for content identification. Renza et al. (2019) use MFCC features with PCA compression, and a Collatz-conjecture-based binarization procedure to obtain a 96-bit code. Shi et al. (2020) construct gammatone filterbank features followed by a random Gaussian projection to derive perceptual speech hashes. Zhang et al. (2018) combine LP-MMSE coefficients with improved spectral entropy to form a binary hash sequence, while Zhang et al. (2021) generate perceptual hashes from the product of sub-band spectrum variance and spectral entropy. As shown in Table 5, these handcrafted schemes struggle to balance the acceptance of benign samples and the rejection of malicious ones. Chromaprint and Zhang et al. (2018) obtain relatively high TPR on VoxCeleb, but at the cost of FPR above 12%, which means that many maliciously tampered samples are incorrectly accepted as benign. In contrast, Renza et al. (2019) and Zhang et al. (2021) maintain lower FPR, but their TPR is reduced and EER is higher, so a noticeable fraction of benign samples is wrongly rejected as tampered. The fingerprint used by SpeeCheck achieves a clearly better operating point on both VoxCeleb and LibriSpeech, with TPR above 98%, AUC above 99.6%, and EER below 2%. These results indicate that the learned fingerprints provide a much clearer separation between benign and malicious modifications, which is what integrity verification requires.

## 5 CONCLUSION

In this paper, we proposed **SpeeCheck**, a proactive and self-contained framework for speech integrity verification. SpeeCheck integrates multiscale feature extraction and contrastive learning to produce robust fingerprints, which are embedded into audio via watermarking. These fingerprints are sensitive to malicious tampering while robust to benign operations commonly introduced during digital distribution, enabling integrity verification without access to external references. Extensive experiments confirm its robustness and sensitivity across diverse tampering scenarios. Notably, evaluations on a constructed real-world dataset further demonstrate its practicality, showing high robustness under social media distribution and strong sensitivity to fine-grained malicious edits.

## ETHICS STATEMENT

This work does not involve human subjects, personally identifiable information, or sensitive data. All experiments are conducted on publicly available datasets (VoxCeleb and LibriSpeech) and a small-scale real-world dataset collected with voluntary consent. To protect privacy, all data used in public demos are anonymized, and no personally identifiable information is released. Deepfake and voice conversion technologies are employed solely to simulate attack scenarios for research evaluation, and no harmful or deceptive content is created or disseminated.

The proposed method aims to strengthen speech integrity verification and mitigate the spread of misinformation. We recognize that, like any integrity verification technology, it could be misused for surveillance or censorship; thus, it should be deployed responsibly and transparently. The authors declare no conflicts of interest or sponsorship-related concerns in this study.

## REPRODUCIBILITY STATEMENT

We make significant efforts to ensure reproducibility. All datasets used in this study are publicly available (VoxCeleb, LibriSpeech), and the constructed real-world dataset is included in the supplementary materials. Details of the fingerprint generation, watermark embedding, training procedure, and evaluation metrics are described in Section 3 and Section 4, with extended information in the Appendix C. An anonymous implementation and demo are provided at `https://speecheck.github.io/SpeeCheck/`, which contains the source code and instructions for reproducing our experiments.

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

## LLM Usage Statement

Large Language Models (LLMs) were used exclusively as general-purpose writing assistants to improve readability and adjust formatting. They did not contribute to the research ideation, methodology, experimental design, analysis, or interpretation of results. All technical content and scientific contributions are solely the work of the authors.

## A  Related Works

### A.1  Passive Detection of Speech Tampering

Audio tampering can introduce detectable inconsistencies in acoustic signals. Traditional passive detection methods rely on statistical or signal-level artifacts introduced during editing. These include frame offset inconsistencies (Yang et al., 2008), local noise level variation (Pan et al., 2012), and interruptions in electric network frequency (Rodríguez et al., 2010; Esquef et al., 2014). Using such patterns, passive detectors are usually implemented as binary classifiers that distinguish authentic from manipulated audio.

With the rise of neural audio generation and deepfake techniques, these handcrafted artifacts become less reliable, since modern synthesis can produce high-quality speech with minimal visible traces. To improve robustness, recent work explores more subtle acoustic features related to fluid dynamics and articulatory phonetics (Blue et al., 2022). Nevertheless, passive detection remains fundamentally limited: it can only observe the received signal, has difficulty generalizing to unseen or adaptive manipulations, and cannot verify whether the speech content originates from the claimed speaker.

### A.2  Proactive Protection of Speech Integrity

Proactive integrity verification embeds or records auxiliary information at publishing time and reuses it during verification. For speech, this is commonly implemented by computing a cryptographic or perceptual hash, or by embedding a watermark payload into the audio signal. At verification time, the received audio is checked by recomputing or extracting this information and comparing it with the expected value.

#### A.2.1  Hash-based methods

Cryptographic hashing, such as SHA256 or MD5, has been widely adopted in industry for integrity verification of digital files (Zakariah et al., 2018). These functions transform a digital audio file into a fixed-length digest, and any bit-level modification changes the digest significantly. This property is ideal for strict file integrity, but it is too sensitive for speech content integrity: common user operations such as format conversion or compression already produce a completely different hash value, which yields a high false alarm rate in realistic audio sharing pipelines.

To reduce this sensitivity, perceptual hashing has been proposed. Instead of operating on raw bytes, perceptual hashes compute content-based digests that are designed to be stable under content-preserving distortions. Typical audio perceptual hashing methods extract features such as cepstral coefficients (Zhang et al., 2021), spectral envelopes (Zhang et al., 2018), or time-frequency energy patterns, and then quantize these features into compact binary codes for matching in a database. These schemes have mainly been developed for copy detection and content retrieval (Microsoft, 2015; Facebook, 2019; Apple, 2021), where robustness to any distortions is more important than distinguishing benign from malicious operations.

More recent work adapts perceptual hashing to integrity verification. For example, Zhang et al. (2021) design an encrypted perceptual hash based on uniform sub-band spectrum variance and spectral entropy of encrypted speech, and Li et al. (2021) propose a reliable audio hash based on Mel-frequency inverted spectrum coefficients and their dynamic parameters. In these methods, handcrafted features are tuned to be robust to benign processing but to change under tampering, and the resulting hash acts as the integrity indicator. However, such handcrafted designs are rigid and may not capture the subtle changes induced by modern generative tampering. In addition, perceptual

hashes still require external storage of reference digests, which is inconvenient in many deployment scenarios.

### A.2.2 FRAGILE AND SEMI-FRAGILE WATERMARKING

Watermarking-based integrity verification follows a different strategy: instead of storing the reference externally, a watermark is directly embedded into the audio signal and later checked for integrity. Fragile watermarking embeds highly sensitive marks whose presence can be destroyed by even minor perturbations (Sripradha & Deepa, 2020; Zhang et al., 2024). This yields strong guarantees that any detected watermark indicates unmodified content, but such schemes are not suitable for everyday audio sharing, where benign operations like compression or resampling already remove or distort the fragile watermark.

Semi-fragile watermarking is designed to survive common benign operations while remaining sensitive to content tampering. For audio, semi-fragile schemes often embed watermark bits in selected hybrid domains (Masmoudi et al., 2020), such as line spectral frequencies (Wang et al., 2019) or wavelet packet subbands (Wang & Fan, 2010), and tune quantization steps and thresholds to balance robustness and fragility for specific attack models. Although these methods can detect a wide range of signal processing operations, they tightly couple the authentication behavior to the watermarking design. As surveyed in (Yu et al., 2017), many new semi-fragile watermarks are proposed only because previous designs fail under newly considered attacks. Supporting new benign operations or tampering types often requires redesigning the embedding rule or the underlying transform.

### A.2.3 COMBINATION OF FINGERPRINT AND WATERMARKING

Another line of work combines content-based fingerprints with watermarking. In these approaches (Gomez et al., 2002; Gulbis et al., 2006; Steinebach & Dittmann, 2003), a compact fingerprint is first extracted from the audio content and then embedded into the signal as a watermark payload. During verification, the embedded fingerprint is extracted and compared with a newly computed fingerprint from the received audio. This self-embedding principle reduces dependence on external databases while still using content-based descriptors for integrity checking.

Early systems in this line use conventional fingerprints and watermark carriers. The fingerprints are derived from handcrafted acoustic features (Seo et al., 2006; 2005) (e.g., subband energies or cepstral-like descriptors) that are tuned to be robust to a predefined set of signal processing operations, and the watermarking schemes are designed for specific channels or codecs. As a result, extending these designs to new manipulation attacks or benign operations is difficult and often requires redesigning both the feature extractor and the embedding rule.

Recently, neural audio watermarking (Chen et al., 2023; Liu et al., 2024a; Roman et al., 2024) has provided a strong carrier for information hiding. Some works have started to use neural watermarking for proactive protection. For example, Ge et al. (2025) propose a proactive defense against speaker identity manipulation by embedding speaker embeddings into speech using audio watermarking. Their method, however, focuses on speaker-identity attacks and does not address semantic content alterations. Other schemes embed payloads that indicate AI-generated or cloned speech (Roman et al., 2024; Liu et al., 2024a), but these payloads are still tailored to specific attack types.

In summary, existing proactive approaches fall into three main categories: hash-based methods, fragile or semi-fragile watermarking, and self-embedding schemes that combine fingerprints with watermarking. Hash-based and perceptual-hash methods are either too sensitive to benign operations or rely on handcrafted features and external hash databases; fragile and semi-fragile watermarking tightly couples robustness and fragility to a fixed embedding design; and prior fingerprint–watermark combinations are built on conventional fingerprints and watermark carriers, or focus on specific attack scenarios. SpeeCheck builds on this line of work by using a learned, operation-selective acoustic fingerprint and a modern neural watermarking scheme within a decoupled architecture, which together enable robust and self-contained integrity verification for both identity and semantic tampering, and can be adapted to new operations by retraining the fingerprint extractor without redesigning the embedding process.

Table 6: Summary of audio operations.

| Operation | Example | Implementation |
|---|---|---|
| ***Benign Operations*** | | |
| Compression | Podcasts, news broadcasts, online meetings | `ffmpeg` (MP3 @ 128 kbps) |
| Reencoding | Saving or uploading audio files | `ffmpeg` (PCM 16-bit) |
| Resampling | Low-bandwidth communication | `Resample (torchaudio)` |
| Noise Suppression | Social media platforms | `RMS-based frame muting` |
| ***Malicious Operations*** | | |
| Deletion | Removing "not" in "I do not agree" | `VAD + remove voiced portion` |
| Splicing | Inserting "not" into "I do agree" | `Insert voiced segment` |
| Substitution | Replacing "agree" with "disagree" | `Swap waveform segment` |
| Silencing | Muting "not" in "I do not agree" | `Mute VAD-detected region` |
| Reordering | Changing sentence order | `Segment + shuffle + concat` |
| Voice Conversion | Changing timbre (speaker identity) | `torchaudio.sox_effects` (training), Voice Changer (testing) |
| Text-to-Speech | Generate new speech with speaker's timbre | `YourTTS (zero-shot synthesis)` |

## B  SPEECHECK DESIGN AND OPERATION DEFINITIONS

### B.1  OVERALL ALGORITHM

The training and verification procedures of SpeeCheck are summarized in Algorithm 1 and Algorithm 2, respectively.

### B.2  DEFINITION OF BENIGN AND MALICIOUS OPERATIONS

We simulate two categories of audio modifications: benign operations and malicious tampering. Benign operations refer to legitimate processing steps encountered during audio storage, transmission, or distribution. These operations do not change the semantic content or the speaker identity of the speech. In contrast, malicious tampering refers to intentional alterations designed to distort either the semantic meaning or the identity of the speaker. We detail each operation below and summarize its characteristics in Table 6.

**Compression.** Lossy compression is applied by converting the waveform to MP3(128 kbps) or AAC (128 kbps), and decoding it back to WAV. This simulates typical processing in podcasts and streaming platforms. We use FFmpeg: `ffmpeg -i input.wav -b:a 128k temp.mp3; ffmpeg -i temp.mp3 output.wav`.

**Reencoding.** The waveform is re-encoded to 16-bit PCM WAV format without compression. This simulates storage or uploading scenarios where minor numerical alterations may occur. Implemented with: `ffmpeg -i input.wav output.wav`.

**Resampling.** Audio is downsampled (e.g., from 16 kHz to 8 kHz) and then upsampled back, simulating low-bandwidth or legacy systems. Implemented with: `torchaudio.transforms.Resample`.

**Noise Suppression.** To simulate automatic noise suppression utilized by social media and streaming platforms, the waveform is divided into overlapping frames. Frames with low root-mean-square (RMS) energy are muted.

---

**Algorithm 1** SpeeCheck Training and Deployment

---

1: **Input:** Raw speech $\mathcal{X}$, benign operations $\mathcal{T}_b(\cdot)$, malicious operations $\mathcal{T}_m(\cdot)$, Wav2Vec2.0 encoder $\varepsilon$, multiscale feature extractor $\mathcal{F}$
2: **Output:** Watermarked speech $\tilde{\mathcal{X}}$
3: **for** $e = 1, 2, \ldots,$ epochs **do**
4:    **for** $b = 1, 2, \ldots,$ batches **do**
5:       $\mathcal{X}^{\text{benign}} \leftarrow \mathcal{T}_b(\mathcal{X}), \quad \mathcal{X}^{\text{malicious}} \leftarrow \mathcal{T}_m(\mathcal{X})$
6:       **Step 1: Frame-level feature extraction**
7:       $\mathcal{Z} \leftarrow \varepsilon(\mathcal{X})$
8:       **Step 2: Multiscale feature summarization**
9:       $\mathbf{h}_n \leftarrow \mathcal{F}(\mathcal{Z})$
10:      **for** $n = 1, \ldots, K$ **do**
11:        $w_n \leftarrow \frac{\exp(\phi(\mathbf{h}_n))}{\sum_{t=1}^{K} \exp(\phi(\mathbf{h}_t))}$
12:      **end for**
13:      $\mathbf{v}' \leftarrow \sum_{n=1}^{K} w_n \cdot \mathbf{h}_n$
14:      $\mathbf{v} \leftarrow \text{Proj}(\mathbf{v}')$
15:      **Step 3: Contrastive fingerprint training**
16:      Compute contrastive loss $\mathcal{L}_c$
17:      Update $\mathcal{F}, \phi,$ Proj via backpropagation
18:    **end for**
19: **end for**
20: **Step 4: Binary fingerprint encoding**
21: $\mathbf{b} \leftarrow \text{sign}(\tanh(\text{Proj}(\text{AttPool}(\mathcal{F}(\varepsilon(\tilde{X}))))))$
22: **Step 5: Segment-wise watermarking**
23: Split $\mathcal{X}$ and $\mathbf{b}$ into $N$ segments: $[\mathcal{X}^{(1)}, \ldots, \mathcal{X}^{(N)}], [\mathbf{b}^{(1)}, \ldots, \mathbf{b}^{(N)}]$
24: **for** $n = 1, \ldots, N$ **do**
25:    $\delta^{(n)} \leftarrow \text{WatermarkEmbedder}(\mathcal{X}^{(n)}, \mathbf{b}^{(n)})$
26:    $\tilde{\mathcal{X}}^{(n)} \leftarrow \mathcal{X}^{(n)} + \delta^{(n)}$
27: **end for**
28: $\tilde{\mathcal{X}} \leftarrow \text{Concat}(\tilde{\mathcal{X}}^{(1)}, \ldots, \tilde{\mathcal{X}}^{(N)})$

---

**Algorithm 2** SpeeCheck Verification

---

1: **Input:** Published speech $\tilde{\mathcal{X}}$, wav2vec2.0 encoder $\varepsilon$, trained multiscale feature extractor $\mathcal{F}$, projection module Proj, attentive pooling AttPool, WatermarkExtractor
2: **Output:** Verification result (Accept or Reject)
3: **Path A: Fingerprint extraction**
4: $\mathbf{b}' \leftarrow \text{sign}(\tanh(\text{Proj}(\text{AttPool}(\mathcal{F}(\varepsilon(\tilde{\mathcal{X}}))))))$
5: **Path B: Segment-wise watermark extraction**
6: Split $\tilde{\mathcal{X}}$ into $N$ segments: $\tilde{\mathcal{X}}^{(1)}, \ldots, \tilde{\mathcal{X}}^{(N)}$
7: **for** $n = 1$ to $N$ **do**
8:    $\hat{\mathbf{b}}^{(n)} \leftarrow \text{WatermarkExtractor}(\tilde{\mathcal{X}}^{(n)})$
9: **end for**
10: $\hat{\mathbf{b}} \leftarrow \text{Concat}(\hat{\mathbf{b}}^{(1)}, \ldots, \hat{\mathbf{b}}^{(N)})$
11: **Integrity decision**
12: **if** $d_H(\mathbf{b}', \hat{\mathbf{b}}) \leq \theta$ **then**
13:
14:    **return** Accept
15: **else**
16:
17:    **return** Reject
18: **end if**

---

**Deletion.**    A portion of speech (not silence) is removed from the speech. For example, deleting "not" from "I do not agree" changes the meaning entirely.

**Splicing.**    A short segment of speech from the same speaker is spliced into the waveform. For example, inserting "not" into the phrase "I do agree" reverses its original semantic meaning.

**Substitution.**    A segment of speech is replaced with another waveform segment of equal length from the same speaker. For instance, replacing "agree" with "disagree" fundamentally changes the intended meaning.

**Silencing.**    A portion of speech (not silence or noise) is deliberately muted by setting its amplitude to zero. For instance, muting the word "not" in "I do not agree" leads to a reversed interpretation.

**Reordering.**    The speech is segmented, rearranged, and concatenated to change the semantic content. For instance, reordering "I never said she stole my money" into "She stole my money, I never said" distorts the original meaning and can lead to an opposite interpretation.

**Voice Conversion.**    Note that integrating voice conversion models into the training pipelines is computationally expensive and time-consuming, making large-scale training impractical. To achieve a comparable effect with lower overhead, during the training phase, we apply pitch shifting for speaker identity modification (e.g., +4 semitones) using SoX effects, implemented via `torchaudio.sox_effects.apply_effects_tensor`. This modification introduces perceptual changes to voice characteristics, effectively creating negative samples for learning to distinguish speaker identity. In the testing phase, we validate SpeeCheck's performance on a separate set of audio manipulated by a state-of-the-art commercial voice changer tool from ElevenLabs (ElevenLabs, 2024).

**Text-to-Speech.**    We synthesize speech from text using a pre-trained text-to-speech (TTS) model, YourTTS (Casanova et al., 2022), which supports multilingual and zero-shot speaker adaptation. This attack can generate speech that closely mimics the speaker's voice with arbitrary semantic content.

**Different Levels of Tampering.**    To evaluate the performance under varying conditions, we define three levels of tampering: minor, moderate, and severe. Specifically, at the minor level, tampering operations, including deletion, splicing, silencing, and substitution, alter about 10% of the original audio content (alteration ratio = 0.1). At the moderate level, these same operations alter 30% of the audio (alteration ratio = 0.3). At the severe level, 50% of the audio is altered (alteration ratio = 0.5), and this level also includes reordering operations, which disrupts the logical structure of the speech.

### B.3    EXPLANATION OF MALICIOUS TAMPERING OVER DIFFERENT GRANULARITIES

Table 7 presents representative examples of malicious tampering at the phoneme, word, and phrase levels. These examples illustrate how manipulations at different temporal granularities can alter the meaning of speech. They also motivate the use of multiscale pooling with window sizes of 20, 50, and 100 frames, which are designed to capture such variations in real-world scenarios.

## C    EXPERIMENTAL SETUP AND EXTENDED RESULTS

### C.1    IMPLEMENTATION DETAILS

To supplement Section 4.1, we provide a detailed description of the model architecture and training configuration.

**Model.** We use the pretrained Wav2Vec2.0 Base model[4] to extract 768-dimensional frame-level acoustic features. These are passed to a two-layer Bidirectional LSTM (BiLSTM) with an input size of 768, a hidden size of 512 (i.e., 256 per direction), and a dropout rate of 0.25. To capture

---

[4]https://github.com/facebookresearch/fairseq/blob/main/examples/wav2vec

Table 7: Examples of malicious tampering at different levels of granularity

| Granularity | Example | Description |
|---|---|---|
| Phoneme-level | Change "bed" to "bad" (English); change "mā" (mother) to "mǎ" (horse) (Mandarin) | Altering a single phoneme can lead to subtle yet meaningful changes. These edits are often difficult to detect but can reverse or distort the intended meaning. |
| Word-level | Insert "not" into "He is guilty" to form "He is not guilty"; replace "approved" with "denied" | Tampering at the word level through insertion, deletion, or substitution can directly modify semantic content, leading to misleading interpretations. |
| Phrase-level | Change "Negotiations will begin immediately" to "Negotiations will be delayed indefinitely" | Reordering or replacing entire phrases can fabricate new narratives while maintaining natural-sounding speech, making the tampering more deceptive. |

temporal features at multiple resolutions, we apply average pooling with window sizes of 20, 50, and 100 frames, with a stride of 10 frames, implemented using `avg_pool1d` along the time axis. The pooled outputs are aggregated by an attentive pooling module consisting of a linear-tanh-linear projection. The resulting weighted sum forms the utterance-level embedding, followed by dropout with a rate of 0.2. This embedding is fed into a two-layer MLP projection head with dimensions $768 \rightarrow 512 \rightarrow 256$, with ReLU activation between layers. The final output vector is L2-normalized and passed through a `tanh` function to constrain values to the range $[-1, 1]$, yielding the continuous-valued fingerprint. For segment-wise watermarking, we use the pretrained AudioSeal model[5] to embed and extract binary fingerprints as watermarks. Each audio is divided into 16 non-overlapping segments, with each segment embedded with a 16-bit binary watermark, resulting in a total payload size of 256 bits per audio.

**Training.** SpeeCheck is trained using a cosine annealing learning rate schedule, decaying from $1 \times 10^{-3}$ to $1 \times 10^{-5}$ over 50 epochs. The contrastive loss is temperature-scaled with $\tau = 0.05$. Training is conducted on 2 NVIDIA A100 GPUs using distributed data parallelism.

Table 8: Examples from RWSID with corresponding editing operations

| Sentence | Editing Operation |
|---|---|
| The board has decided they can not approve the new budget. | Deletion / Silencing ("not") |
| Our analysis shows this investment is not a secure option. | Deletion / Silencing ("not") |
| Based on the evidence, the suspect is innocent. | Substitution → "guilty" |
| Based on the evidence, the suspect is guilty. | Substitution → "innocent" |
| I never said she stole the company's data. | Reordering |
| I never said she stole the company's data. | Voice Conversion (AI) |
| We will begin the product launch immediately. | Replacement → "delay" |
| We will delay the product launch immediately. | Replacement → "begin" |
| I believe it is a good idea, but we need more time. | Splicing |
| This is authentic audio, not deepfake. | Text-to-Speech (AI) |

## C.2 DATASET AND EVALUATION DETAILS

We use two public speech datasets: **VoxCeleb** and **LibriSpeech**. For VoxCeleb, the development set is used for training and the test set for evaluation. For LibriSpeech, we use only the `test-clean` subset for evaluation. To comprehensively evaluate the effectiveness of SpeeCheck in real-world scenarios, we construct a **Real-World Speech Integrity Dataset** (RWSID). This dataset comprises recordings from 10 volunteers with diverse demographic backgrounds (including multiple races and sexes). Each participant read 8 prepared speeches (see Table 8). All audio files are converted to WAV format and resampled to 16 kHz.

---

[5]https://github.com/facebookresearch/audioseal

**Preprocessing.** We randomly sample 10,000 utterances from the VoxCeleb development set for model training. For evaluation, we sample 500 utterances each from the VoxCeleb test set and the LibriSpeech `test-clean` subset. To stabilize the training and ensure data quality, we retain only utterances with durations between 2 and 20 seconds. We further analyze the effect of utterance duration in Appendix C.7.

For each valid utterance, we generate two sets of augmented variants for contrastive learning: (i) Benign Augmentations: These are modifications that preserve both speaker identity and semantic content. (ii) Malicious Augmentations: These include tampering operations intended to alter speaker identity and semantic content. The details can be found in Appendix B.2.

**Evaluation Metrics.** For evaluation, we consider benign and malicious as positive and negative classes, respectively. TP is the number of benign samples correctly classified, and FN is the number of benign samples incorrectly classified as malicious. FP is the number of malicious samples incorrectly classified as benign, and TN is the number of malicious samples correctly rejected. The following metrics are computed:

- True Positive Rate (TPR): $\text{TPR} = \text{TP}/(\text{TP} + \text{FN})$
- False Positive Rate (FPR): $\text{FPR} = \text{FP}/(\text{FP} + \text{TN})$
- True Negative Rate (TNR): $\text{TNR} = \text{TN}/(\text{TN} + \text{FP})$
- False Negative Rate (FNR): $\text{FNR} = \text{FN}/(\text{FN} + \text{TP})$
- Equal Error Rate (EER): The error rate at the decision threshold where FPR = FNR.
- Area Under the Curve (AUC): The area under the receiver operating characteristic curve.

### C.3 SIMILARITY DISTRIBUTION USING MFCC FEATURE

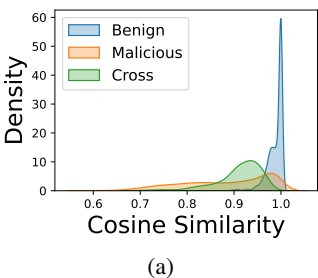 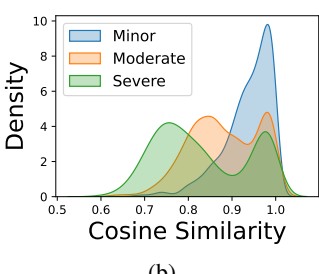

(a)              (b)

Figure 6: Probability distributions: (a) MFCC embedding similarity to original audio under different modifications; (b) MFCC embedding similarity to original audio at different tampering levels.

To complement the observations in Section 2.3, we present similarity distributions computed using handcrafted Mel-frequency cepstral coefficients (MFCC) instead of wav2vec2 embeddings. As shown in Figure 6, the similarity distributions between original and modified audio samples using MFCC features exhibit trends similar to those observed with wav2vec2-based representations. Specifically, the distributions corresponding to benign and malicious modifications overlap, and the similarity scores tend to decrease as the extent of tampering increases. This indicates that MFCC-based similarity comparison can only measure the extent of modification but does not effectively distinguish between different types of modifications.

### C.4 EVALUATION ON SEMANTIC AND IDENTITY CHANGES UNDER BENIGN AND MALICIOUS OPERATIONS

We evaluate the impact of different audio modifications on both semantic integrity and speaker identity consistency. Semantic preservation is quantified using word error rate (WER) computed from a pre-trained automatic speech recognition (ASR) model[6], `facebook/wav2vec2-base-960h`,

---

[6]`https://github.com/facebookresearch/fairseq/blob/main/examples/wav2vec`

Table 9: WER and Identity Similarity under Different Operations

| Operation | WER % | Identity Similarity % |
|---|---|---|
| ***Benign Operations*** | | |
| Compression | 1.15 | 95.60 |
| Reencoding | 0.26 | 99.99 |
| Resampling | 6.87 | 78.00 |
| Noise suppression | 8.24 | 94.52 |
| ***Malicious Operations*** | | |
| Deletion (minor) | 21.65 | 99.04 |
| Deletion (moderate) | 40.20 | 96.84 |
| Deletion (severe) | 62.32 | 93.29 |
| Splicing (minor) | 31.24 | 99.00 |
| Splicing (moderate) | 52.45 | 97.35 |
| Splicing (severe) | 78.36 | 96.55 |
| Silencing (minor) | 30.76 | 98.16 |
| Silencing (moderate) | 53.79 | 90.13 |
| Silencing (severe) | 75.74 | 75.96 |
| Substitution (minor) | 23.22 | 98.33 |
| Substitution (moderate) | 48.12 | 94.36 |
| Substitution (severe) | 63.03 | 90.72 |
| Reordering | 69.55 | 99.53 |
| Text-to-speech | - | - |
| Voice conversion | 8.60 | **41.60** |

a CTC-based ASR model. Speaker identity preservation is measured by cosine similarity between embeddings extracted using the pre-trained speaker verification (SV) model[7], `speechbrain/spkrec-ecapa-voxceleb`.

As shown in Table 9, benign operations (e.g., compression, reencoding, resampling, noise suppression) result in low WER ($\leq 8.24\%$) and high identity similarity ($\geq 78\%$), indicating that they largely preserve both semantic content and speaker identity. In contrast, malicious operations introduce substantial degradation. WER increases steadily with the severity of deletion, splicing, silencing, and substitution, reflecting significant semantic changes. These operations, however, generally maintain high identity similarity because they retain the original timbre. Notably, voice conversion results in relatively low WER, but significantly reduces identity similarity (41.60%), since it deliberately alters the speaker's timbre.

To further investigate the nonzero WER observed under benign operations, we manually examined the ASR outputs. Most transcription errors were minor substitutions or alignment shifts that did not affect the overall meaning. This suggests that the observed WER in these cases reflects limitations of the ASR model and metric sensitivity rather than genuine semantic distortion.

## C.5 RESULTS OF FINE-GRAINED MALICIOUS OPERATIONS REJECTION

We report the detection performance of SpeeCheck on fine-grained malicious operations across varying degrees of tampering severity, as shown in Table 10.

## C.6 EVALUATION IN REAL-WORLD SCENARIO

To validate SpeeCheck's performance in practical settings, we conducted evaluations on the RWSID dataset (described in Appendix C.2). Example recordings and verification results are available on our demo page.[8] We then designed two evaluation scenarios to simulate real-world challenges:

---

[7]https://huggingface.co/speechbrain/spkrec-ecapa-voxceleb
[8]https://speecheck.github.io/SpeeCheck/

Table 10: Results of fine-grained malicious operation rejection on VoxCeleb and LibriSpeech.

| Operation | VoxCeleb | | | | LibriSpeech | | | | Semantic | Identity |
|---|---|---|---|---|---|---|---|---|---|---|
| | TNR | FNR | AUC | EER | TNR | FNR | AUC | EER | | |
| Deletion (minor) | 100.00 | 1.01 | 100.00 | 0.00 | 100.00 | 1.01 | 99.92 | 0.20 | ✗ | ✓ |
| Deletion (moderate) | 100.00 | 1.61 | 100.00 | 0.00 | 100.00 | 1.81 | 100.00 | 0.00 | ✗ | ✓ |
| Deletion (severe) | 100.00 | 1.01 | 100.00 | 0.00 | 100.00 | 1.81 | 100.00 | 0.00 | ✗ | ✓ |
| Splicing (minor) | 100.00 | 0.80 | 100.00 | 0.00 | 100.00 | 1.61 | 99.99 | 0.40 | ✗ | ✓ |
| Splicing (moderate) | 100.00 | 0.20 | 100.00 | 0.00 | 100.00 | 2.21 | 100.00 | 0.00 | ✗ | ✓ |
| Splicing (severe) | 100.00 | 1.01 | 100.00 | 0.00 | 100.00 | 1.41 | 99.98 | 0.10 | ✗ | ✓ |
| Silencing (minor) | 97.59 | 0.80 | 99.44 | 1.71 | 95.57 | 1.41 | 99.52 | 3.22 | ✗ | ✓ |
| Silencing (moderate) | 98.99 | 0.00 | 99.68 | 0.91 | 99.40 | 1.61 | 99.92 | 0.91 | ✗ | ✓ |
| Silencing (severe) | 99.20 | 1.41 | 99.81 | 1.11 | 99.40 | 2.41 | 99.68 | 1.51 | ✗ | ✓ |
| Substitution (minor) | 93.36 | 1.01 | 99.43 | 2.92 | 79.88 | 1.81 | 97.79 | 8.35 | ✗ | ✓ |
| Substitution (moderate) | 100.00 | 0.80 | 99.99 | 0.40 | 99.20 | 1.81 | 99.46 | 1.51 | ✗ | ✓ |
| Substitution (severe) | 100.00 | 0.80 | 100.00 | 0.00 | 100.00 | 1.81 | 99.84 | 1.31 | ✗ | ✓ |
| Reordering | 97.59 | 0.60 | 98.62 | 2.11 | 98.39 | 1.41 | 99.21 | 1.71 | ✗ | ✓ |
| Text-to-speech | 100.00 | 0.00 | 100.00 | 0.00 | 100.00 | 0.00 | 100.00 | 0.00 | ✗ | ✓ |
| Voice conversion | 99.40 | 0.00 | 100.00 | 0.00 | 97.80 | 0.00 | 100.00 | 0.00 | ✓ | ✗ |
| **Overall** | 99.08 | 0.74 | 99.80 | 0.61 | 97.98 | 1.48 | 99.69 | 1.28 | - | - |

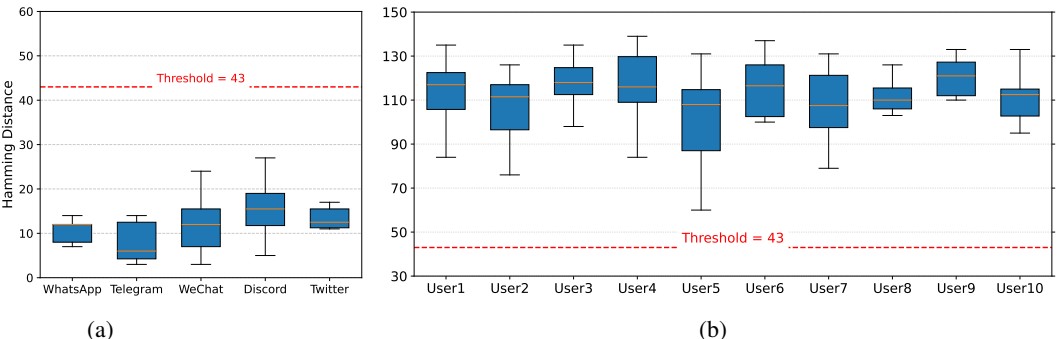

(a)                    (b)

Figure 7: Hamming distance distributions for real-world scenarios: (a) benign social media distribution and (b) malicious tampering.

**Benign Distribution.** To assess SpeeCheck's robustness under practical distribution scenarios, the protected audios were uploaded to widely used social media platforms (including WhatsApp, Telegram, WeChat, Discord, and Twitter) and subsequently downloaded after platform-side preprocessing. These steps reflect realistic distribution pipelines where compression, reencoding, and noise suppression may be applied.

**Malicious Tampering.**: To evaluate its sensitivity to sophisticated attacks, we performed fine-grained edits manually, including deletion, splicing, silencing, substitution, and reordering. Moreover, we tested commercial platforms for voice conversion (ElevenLabs, 2024) and text-to-speech synthesis (Vocloner, 2024).

Figure 7a shows that the Hamming distances of audios redistributed via social media consistently remain below the detection threshold, confirming SpeeCheck's robustness against real-world distribution. In contrast, Figure 7b demonstrates that all malicious edits yield Hamming distances above the threshold across all 10 users, indicating its reliable sensitivity to real-world tampering.

## C.7 EVALUATION ON DIFFERENT LENGTHS OF SPEECH

To evaluate SpeeCheck on longer audio recordings, we test speech samples with durations ranging from 20 seconds to 10 minutes. As shown in Table 11, the system performs well across all lengths. While performance gradually degrades with increasing duration, the EER rises from 1.57% at 20 seconds to 8.41% at 10 minutes, the overall detection remains robust, with the AUC consistently above 96%.

This behavior is consistent with the roles of the two main components in SpeeCheck. The acoustic fingerprint summarizes the entire segment into a single 256-bit representation. When the segment becomes very long, local manipulations (for example, deleting or substituting a single word) affect only a small portion of the frames, so their influence on the global fingerprint can be diluted, which increases the EER. At the same time, the watermarking backbone (AudioSeal) acts only as a carrier for this fingerprint. If segments were extremely short, the effective embedding capacity would be limited and watermark extraction on benign audio could become unstable, which would increase false rejections. The results in Table 11 show that, within the tested range, SpeeCheck maintains a good balance between these two effects.

Table 11: Performance of SpeeCheck under different speech durations.

| Speech Duration | TPR | FPR | AUC | EER |
|---|---|---|---|---|
| 20s | 98.75 | 1.65 | 99.69 | 1.57 |
| 60s | 92.50 | 4.88 | 98.39 | 5.87 |
| 5m | 94.87 | 7.05 | 98.02 | 6.09 |
| 10m | 90.95 | 7.76 | 96.04 | 8.41 |

## C.8 EVALUATION ON DIFFERENT LANGUAGES OF SPEECH

SpeeCheck operates directly on acoustic features rather than on linguistic semantic content. Therefore, language differences should not interfere with the fingerprint generation process. To empirically validate this, we conduct an additional evaluation on the multilingual FLEURS dataset (Conneau et al., 2023), which includes 102 languages. We select six representative languages (French, German, Chinese, Spanish, Japanese, and Polish) and evaluate SpeeCheck under the same protocol as in the main experiments, including the same benign operations and tampering attacks.

As shown in Table 12, SpeeCheck maintains high robustness to benign operations and high sensitivity to tampering across all six languages. All AUC scores are above 98%, and the EER remains below 8%, which is comparable to the results on the English-centric datasets in the main paper.

Table 12: Performance of SpeeCheck on different speech languages.

| Language | TPR | FPR | AUC | EER |
|---|---|---|---|---|
| French | 98.27 | 3.37 | 99.54 | 2.47 |
| German | 98.62 | 2.20 | 99.61 | 2.03 |
| Chinese | 96.70 | 3.12 | 99.37 | 3.21 |
| Spanish | 99.69 | 1.78 | 99.83 | 1.07 |
| Japanese | 92.27 | 1.80 | 98.47 | 7.09 |
| Polish | 98.37 | 8.06 | 98.02 | 7.06 |

## C.9 REAL-TIME EVALUATION

Table 13: Real-time performance of SpeeCheck.

| Process | Real-Time Coefficient (RTC) |
|---|---|
| Protection | 0.02× |
| Verification | 0.03× |

We evaluate computational efficiency using the Real-Time Coefficient (RTC), defined as the ratio of processing time to audio duration. As shown in Table 13, both protection and verification achieve RTC values well below 1×, confirming the practicality of SpeeCheck for real-time use.

## C.10 WATERMARKED SPEECH QUALITY

We evaluate the perceptual quality of watermarked speech using four objective metrics. (1) Scale-Invariant Signal to Noise Ratio (SI-SNR) quantifies waveform-level distortion in decibels (dB). Higher values indicate less distortion. (2) Perceptual Evaluation of Speech Quality (PESQ) (Rix et al., 2001) ranges from 1.0 (poor) to 4.5 (excellent), and reflects perceived speech quality. (3) Short-Time Objective Intelligibility (STOI) (Taal et al., 2010) ranges from 0 to 1, with higher values indicating better intelligibility. (4) Log Spectral Distance (LSD) measures spectral deviation between original and watermarked speech, lower values indicate greater spectral fidelity.

As shown in Table 14, our proposed SpeeCheck has little perceptual degradation. The high SI-SNR and PESQ scores, along with near-perfect intelligibility (STOI) and low spectral error (LSD), demonstrate that the watermarking process preserves both fidelity and intelligibility, making it suitable for practical deployment.

Table 14: Audio quality metrics.

| Methods | SI-SNR | PESQ | STOI | LSD |
|---|---|---|---|---|
| SpeeCheck | 25.14 | 4.28 | 0.998 | 0.111 |

## C.11 ABLATION STUDIES

To assess the contribution of SpeeCheck's core modules, we conduct ablation studies at two levels. Unless otherwise stated, all ablations are performed on the VoxCeleb1 test set. At the system level, we ablate three key components: the multiscale feature extractor, the attentive pooling module, and the contrastive learning objective. We evaluate the multiscale feature extractor by removing the multiscale branch and using the direct output of the BiLSTM as the fingerprint feature. For temporal pooling, we substitute attentive pooling with average pooling. Finally, we compare the InfoNCE loss (Oord et al., 2018) with the widely used Triplet Loss (Schroff et al., 2015). As shown in Table 15, each module contributes to the overall performance. Removing the multiscale feature extractor leads to a significant degradation, indicating the importance of capturing both global and local temporal patterns. Substituting attentive pooling with average pooling reduces performance, indicating that the attention mechanism provides better frame selection for embedding generation. Replacing InfoNCE with Triplet Loss causes a substantial performance decline, showing that InfoNCE is more effective for learning discriminative embeddings in our task.

Table 15: Ablation study on feature extractor, temporal pooling scheme, and loss function.

| Method Variant | TPR | FPR | AUC | EER |
|---|---|---|---|---|
| SpeeCheck (Multiscale → w/o Multiscale) | 94.48 | 5.26 | 98.50 | 5.29 |
| SpeeCheck (AttentivePooling → AvgPooling) | 93.80 | 6.25 | 98.42 | 6.22 |
| SpeeCheck (InfoNCE Loss → Triplet Loss) | 85.94 | 1.46 | 95.77 | 10.73 |
| **SpeeCheck** | **99.14** | **0.80** | **99.85** | **0.83** |

Since the multiscale feature extractor is the main component of the fingerprint generation module, we further provide a more detailed evaluation of this part. Specifically, we instantiate a variant of SpeeCheck where the multiscale branch is removed and only the BiLSTM output is used as the fingerprint feature, while keeping the training protocol, datasets, and evaluation metrics identical to the main experiments. The detailed per-operation results of this variant are reported in Tables 16 and 17. Compared to the full model, the overall performance drops for both benign and malicious operations, and the degradation is particularly clear for subtle tampering such as Substitution (minor), which confirms that the multiscale design is crucial for the fingerprint generator.

Table 16: Benign-operation performance of the variant without the multiscale module.

| Operation | TPR | FPR | AUC | EER |
|---|---|---|---|---|
| Compression | 79.60 | 13.40 | 90.41 | 14.40 |
| Reencoding | 100.00 | 10.20 | 99.85 | 0.90 |
| Resampling | 99.40 | 11.00 | 99.52 | 3.20 |
| Noise suppression | 100.00 | 11.00 | 99.84 | 0.70 |
| **Overall** | 94.75 | 11.40 | 97.41 | 4.80 |

Table 17: Tampering-detection performance of the variant without the multiscale module.

| Operation | TNR | FNR | AUC | EER |
|---|---|---|---|---|
| Deletion (minor) | 83.60 | 6.00 | 97.53 | 9.30 |
| Deletion (moderate) | 97.40 | 5.60 | 99.22 | 4.00 |
| Deletion (severe) | 99.40 | 4.60 | 99.68 | 1.60 |
| Splicing (minor) | 96.00 | 4.00 | 99.16 | 4.00 |
| Splicing (moderate) | 98.60 | 4.80 | 99.60 | 2.70 |
| Splicing (severe) | 99.60 | 6.00 | 99.83 | 2.10 |
| Silencing (minor) | 79.60 | 5.20 | 96.37 | 13.70 |
| Silencing (moderate) | 82.00 | 5.20 | 97.00 | 11.60 |
| Silencing (severe) | 85.80 | 4.00 | 97.80 | 9.60 |
| Substitution (minor) | 64.60 | 5.40 | 92.55 | 20.80 |
| Substitution (moderate) | 70.60 | 5.60 | 94.45 | 18.10 |
| Substitution (severe) | 81.00 | 4.20 | 96.74 | 13.00 |
| Reordering | 93.20 | 6.60 | 97.85 | 6.70 |
| Text-to-speech | 100.00 | 0.00 | 100.00 | 0.00 |
| Voice conversion | 100.00 | 4.80 | 99.75 | 2.40 |
| **Overall** | 87.55 | 4.74 | 98.09 | 7.97 |

### C.12 EFFECT OF WATERMARKING ON FINGERPRINT STABILITY

To verify that watermarking does not destroy the acoustic fingerprint that it is supposed to protect, we compare fingerprints before and after watermark embedding. For each audio sample, we compute a binary fingerprint from the original audio and from its watermarked version, and measure the Hamming distance between these two. Table 18 summarizes the average distances on three datasets. The average Hamming distances are 6.87 on VoxCeleb, 9.15 on LibriSpeech, and 5.40 on the RWSID dataset, for a fingerprint length of 256 bits and a decision threshold $\theta = 42$. These values are much smaller than the threshold, showing that watermarking introduces only minor variations to the fingerprint and therefore does not affect the integrity verification decision.

Table 18: Hamming distance between fingerprints before and after watermark embedding.

| Dataset | VoxCeleb | LibriSpeech | RWSID |
|---|---|---|---|
| Average Hamming Distance | 6.87 | 9.15 | 5.40 |

### C.13 ERROR ANALYSIS OF FALSE POSITIVES AND FALSE NEGATIVES

As discussed in Section 3.3, the integrity verification pipeline in SpeeCheck contains two paths: fingerprint recalculation and embedded fingerprint extraction. From the received audio, Path A recomputes an acoustic fingerprint $\mathbf{f}_A$, while Path B decodes the embedded fingerprint $\mathbf{f}_B$ from the watermark carrier. The decision is based on the Hamming distance between these two fingerprints: we predict benign if $d_H(\mathbf{f}_A, \mathbf{f}_B) \leq \theta$ and tampered otherwise, where $\theta$ is the decision threshold. We recall that benign speech is treated as the positive class and tampered speech as the negative class in all experiments.

A False Positive (FP) corresponds to a tampered sample that is incorrectly accepted as benign. In our system, this happens when a malicious edit is subtle enough that $\mathbf{f}_A$ does not change sufficiently, so that $d_H(\mathbf{f}_A, \mathbf{f}_B)$ remains below $\theta$. In these cases, the watermark decoder in Path B still reliably recovers the original embedded fingerprint, and the error is mainly due to the limited sensitivity of the recomputed fingerprint. This effect is most visible in the "Substitution (minor)" category (see Table 10), where the TNR is lower than in other tampering categories.

A False Negative (FN) corresponds to a benign sample that is incorrectly rejected. We observe two main causes for such errors. In some benign cases, aggressive but allowed operations (for example, strong compression or noise suppression) can perturb the acoustics so much that $\mathbf{f}_A$ drifts far from $\mathbf{f}_B$ and the distance exceeds the threshold, leading to an unnecessary rejection. In a smaller number of cases, the watermarking carrier (AudioSeal) may fail to decode a stable fingerprint, and $\mathbf{f}_B$ contains many bit errors, which also increases the distance. Since the carrier is modular, future work can replace AudioSeal with a more robust watermarking scheme to further reduce this type of failure.

### C.14 HANDLING BORDERLINE CASES IN DEPLOYMENT

In real deployments, the small fraction of remaining failure cases can be handled through a simple hierarchical verification strategy. Instead of using a single hard threshold $\theta$, the system can define a narrow uncertainty band around this value, with two thresholds $\theta_{\text{low}} < \theta_{\text{high}}$.

Samples with Hamming distance $d_H(\mathbf{b}, \mathbf{b}') \leq \theta_{\text{low}}$ are accepted as benign, and samples with $d_H(\mathbf{b}, \mathbf{b}') \geq \theta_{\text{high}}$ are rejected as tampered. Only samples that fall into the uncertainty interval $(\theta_{\text{low}}, \theta_{\text{high}})$ are flagged as borderline cases and sent to secondary checks, such as human review or additional forensic tools, depending on the deployment. This design does not change the core SpeeCheck pipeline, but it provides a practical way to handle rare edge cases in a controlled and predictable manner.

### C.15 VISUALIZATION OF MULTISCALE FEATURE EXTRACTION

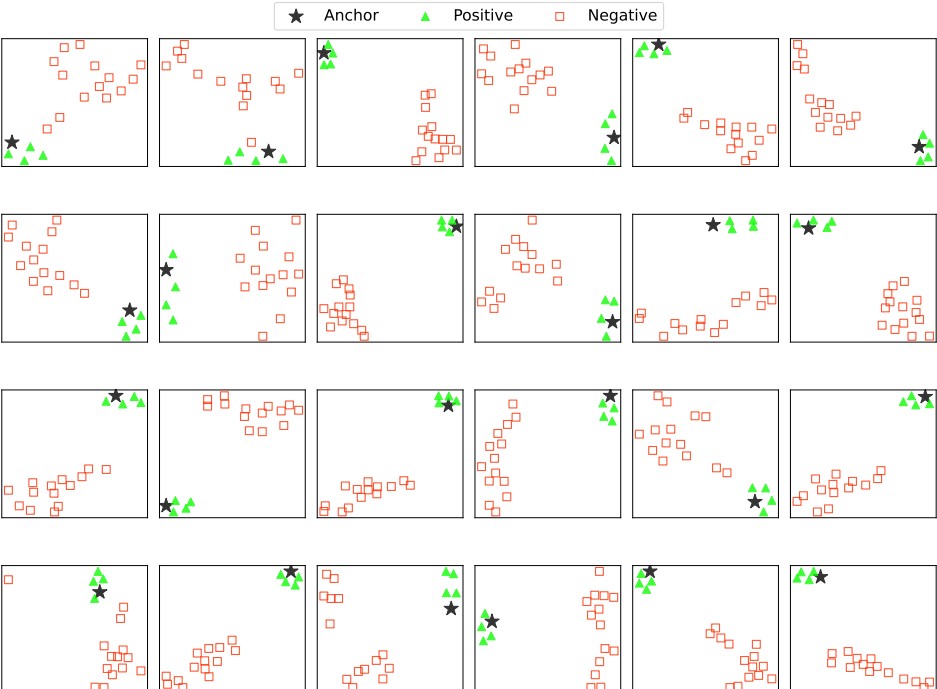

Figure 8: t-SNE visualizations of speech samples (after training).

# D   SECURITY DISCUSSION AND EXTENSIONS

## D.1   ADVERSARIAL ATTACKS ON DEEPFAKE DETECTORS

Liu et al. (Liu et al., 2024b) evaluate strong targeted adversarial attacks on deepfake speech detectors, including PGD(Madry et al., 2017), I-FGSM (Kurakin et al., 2018), and SimBA (Guo et al., 2019). These methods are designed for passive detectors such as RawNet2 (Tak et al., 2021) and AASIST (Jung et al., 2022), which treat deepfake detection as a binary classification problem and learn a decision boundary that separates "Real" from "Fake" based on synthesis artifacts. The adversarial attacks work by adding small, imperceptible perturbations so that a given fake sample is pushed across this decision boundary and is misclassified as "Real."

SpeeCheck follows a different verification principle. Instead of deciding authenticity from artifacts alone, it checks the consistency between two quantities: (i) a 256-bit fingerprint that has been embedded into the audio via watermarking, and (ii) a fingerprint recomputed from the received audio using the same feature extractor. Deepfake audio generated from scratch does not contain an embedded fingerprint that matches this verification protocol. Under the threat model of (Liu et al., 2024b), an attacker may add adversarial noise to a deepfake sample, but such an additive perturbation cannot by itself create a valid 256-bit fingerprint that is consistent with the internal SpeeCheck pipeline. As a result, the adversarially perturbed deepfake still fails the fingerprint–watermark consistency check and is rejected.

Formally analyzing adversarial robustness against adaptive attackers who explicitly target the fingerprint–watermark verification mechanism is an important direction for future work. However, the above discussion shows that SpeeCheck is not directly vulnerable to the same class of adversarial examples studied in (Liu et al., 2024b), because it does not rely on a single artifact-based classification boundary.

## D.2   SECURITY EXTENSION AGAINST REPLAY ATTACKS

In practical deployments, if the feature extractor and watermarking pipeline are publicly available, an attacker could manipulate the audio content, run the same feature extractor as the verifier to compute a continuous fingerprint $\mathbf{v}$, and then re-embed this fingerprint into the audio stream using the watermarking scheme. Such a replay-style attack could produce forged audio that passes the verification. To mitigate this risk, SpeeCheck can be extended with a simple secret-key mechanism in the fingerprint binarization step, inspired by biohashing and similarity-preserving hashing (Jin et al., 2004; Charikar, 2002; Evennou et al., 2025).

In the basic design, the final binary fingerprint is obtained by applying a sign function to the continuous vector $\mathbf{v} \in \mathbb{R}^{d_v}$,

$$\mathbf{b} = \mathrm{sign}(\mathbf{v}), \tag{3}$$

where $\mathbf{b} \in \{-1, +1\}^d$ is the binary code used for integrity verification. To secure this step, we replace the direct binarization with a keyed projection:

$$\mathbf{b}_{\mathrm{sec}} = \mathrm{sign}(\mathbf{v}\mathbf{R}), \tag{4}$$

where $\mathbf{R} \in \mathbb{R}^{d_v \times d_v}$ is a secret orthogonal matrix that acts as a private key shared between the embedder and the verifier. Only $\mathbf{b}_{\mathrm{sec}}$ is embedded and later recovered for verification; the matrix $\mathbf{R}$ is never exposed to the adversary.

This modification brings two advantages. First, an attacker who only observes the public feature extractor cannot forge a valid binary code. Even if they can compute $\mathbf{v}$ from a manipulated audio, they do not know $\mathbf{R}$ and thus cannot construct the correct secured fingerprint $\mathbf{b}_{\mathrm{sec}}$ that matches the verifier's output. The non-linear sign function further discards magnitude information and makes it difficult to infer $\mathbf{R}$ from observed binary codes. Second, because $\mathbf{R}$ is orthogonal, it preserves distances in the continuous feature space. As a result, the robustness properties of SpeeCheck are

maintained: benign operations and malicious tampering still induce similar distance patterns after the keyed transform, so the decision rule based on the Hamming distance remains effective.

In practice, the secret matrix $\mathbf{R}$ can be derived from a shorter cryptographic key using a pseudo-random generator and refreshed when necessary. In many realistic deployments, the fingerprint generator and the watermark embedder would also be provided as managed services rather than released as public models, which further increases the practical difficulty for attackers, although we do not rely on obscurity as a formal security guarantee. A complete formal security analysis of this extension is left for future work.

## E    DISCUSSION

While SpeeCheck provides a new paradigm for self-contained speech integrity verification, we acknowledge several limitations that can be improved in future research: 1) SpeeCheck's robustness is limited concerning certain operations like time-stretching (speeding up/slowing down) and Voice Activity Detection (VAD). These operations are sometimes not malicious, but they can inherently alter pitch, tempo, or meaningful phonemes in the speech. Our framework prioritizes a security-first design, we conservatively treat modified speech as unreliable. But extending training with such operations, or integrating more robust watermarking schemes, could improve the applicability in the future. 2) SpeeCheck is optimized for speech durations between 2 and 20 seconds. For very short clips, the embedding and watermark extraction may become unstable. However, such clips often lack meaningful semantic content, making them less critical targets for tampering. For very long audio, although we did not explicitly train on durations beyond 20 seconds, SpeeCheck still exhibited reasonable generalization. A practical solution is to segment longer recordings into multiple 20-second chunks, protect and verify the content in controllable chunks. 3) Current SpeeCheck is tailored for speech integrity verification rather than general audio (e.g., music, environmental sounds). This choice is motivated by the fact that Speech is particularly vulnerable to tampering and can significantly impact social trust and social stability. Generalizing the system to broader audio domains would be a promising direction for future work, for instance, emphasizing perceptual fidelity, spectral consistency, and artistic style preservation. 4) Current SpeeCheck design only provides an utterance-level integrity decision without explicitly localizing the tampered region. While this is sufficient for the targeted use cases of general users and platforms who mainly need a clear "authentic vs. tampered" verdict per utterance, finer-grained temporal localization would be valuable for forensic analysts. Extending SpeeCheck with segment-level localization capabilities is an interesting direction for future work.

