# OpenReview forum: "SpeeCheck: Self-Contained Speech Integrity Verification via Embedded Acoustic Fingerprints"
_ICLR.cc/2026/Conference — Submitted to ICLR 2026_

### Official Review · Reviewer_JyhN · 2025-10-21

**Soundness:** 2
**Presentation:** 2
**Contribution:** 2
**Rating:** 2
**Confidence:** 4

**Summary:**

The submission proposes a mechanism for detecting manipulated voice speech.
First, a feature extraction is trained via constrative learning to be discriminative:
- robust to bening audio eidting
- fragile to malicious audio editing.

Second, this feature is embedded in the audio stream thanks to a watermarking technique.

At detection time, one compares the features decoded from the watermark with the feature extracted from the audio.
A large difference means that the audio has been maliciously modified.

**Strengths:**

A nice assemblage of different technology bricks.

**Weaknesses:**

### W0 Lack of bibliographic references

Embedding a hash/perceptual hash within the content via a watermarking technique is a very old idea. It pertains to the so-called semi-fragile watermarking schemes dedicated to content authentication. See, for instance, the references in the overview *Review on Semi-Fragile Watermarking Algorithms for Content Authentication of Digital Images*, Yu et al., MDPI Future Internet.

As a consequence, I challenge the first two contributions. Accessing the original speech recordings has never been considered as an option in the literature. Using watermarking to hide a signature/fingerprint/perceptual hash/feature (whatever the name) is not new.
It appears to me that the primary contribution is the multiscale features sensitive to malicious manipulations.


### W1 Remaining issues

Moreover, this literature usually elaborates on three problems:
- Watermarking should not destroy the perceptual hash. No consideration of this topic is given here. More broadly, it would be good to discover the reason behind false negatives: is it due to a watermarking decoding failure or due to the feature extraction failure?
Localization: This literature not only decides whether the content is authentic but also spots where the manipulation happened within some granularity. There is no localization in this submission.
- Security. The audio watermarking technique AudioSeal is on the shelves, The proposed feature extraction might be as well. Therefore, there is no secret key. An attack can manipulate the content, recalculate the new feature and re-embed it in the audio stream. No consideration about the security in the submission.

### W2 Benchmark

- Since the feature extraction is the main contribution, a dedicated benchmark is welcome.
- As for authentication, AudioSeal can hide a 16-bit message, but its decoder also gives a detection score per sample. The absence of watermark signifies the manipulation. A comparison with AudioSeal is definitely missing. (However, I assume here that one can learn a private AudioSeal)

### W3 Naive argumentation

The motivation summarized in Figure 2 is naive. No professional would ever consider SHA256 hash as a suitable feature representation for multimedia content. In the same way, Table 4 is obvious: deepfakes are not watermarked.

### W4 Experimental results
I have some concerns about the evaluation metric (paragraph Line 337). Note that the threshold is fixed.
- How can you compute an AUC if the threshold is fixed? Anyway, AUC is a really bad metric for applications requiring small FPR.
- How can you compute an EER if the threshold is fixed?
- Why is the FPR varying in Table 1? Table 1 deals only with positive (ie. benign) content. Therefore, there is no way to measure a FPR as this requires negative (ie. manipulated) content. Moreover, the fact that the FPR varies from one benign operation to another can not be correct. The same happens with Table 2 where FNR evaluations are wrong. Also, note that, in principle,  FNR + TPR = 100 and FPR + TNR = 100, which is not the case here. Conclusion: Table 1 should report only TPR and Table 2 only FPR.

**Questions:**

### Q1

Since most of the operations deal with the semantics, why not hiding the transcript Speech-to-Text inside the audio stream with the help of LLM zip. Like this paper did for images: *FAST, SECURE, AND HIGH-CAPACITY IMAGE WATERMARKING WITH AUTOENCODED TEXT VECTORS*, Evennou et al.

### Q2

Why the TNR is lower than 100 for the voice conversion operation? As for Text-to-Speech. Isn't it a new audio stream not-watermarked?

### Q3

Embedding rate. The 256-dimension feature is embedded into 16 segments. How many seconds does this represent? What happen next if the audio stream is longer? Do you extract and embed another feature or is it always the same "global" feature which is embedded repeatedly?

---

> ### Author Response · Authors · 2025-11-21
> **Author Response (1/n)**
>
> Thank you for providing your valuable and constructive feedback! Please find below the responses to each comment.
>
> ---
>
> > **[Q1] Embedding a hash/perceptual hash within the content via a watermarking technique is a very old idea. It pertains to the so-called semi-fragile watermarking schemes dedicated to content authentication. See, for instance, the references in the overview Review on Semi-Fragile Watermarking Algorithms for Content Authentication of Digital Images, Yu et al., MDPI Future Internet. As a consequence, I challenge the first two contributions. Accessing the original speech recordings has never been considered as an option in the literature. Using watermarking to hide a signature/fingerprint/perceptual hash/feature (whatever the name) is not new. It appears to me that the primary contribution is the multiscale features sensitive to malicious manipulations.**
>
> Thank you for this valuable comment.
>
> We fully agree that embedding a hash into the audio via watermarking for authentication is a well-established concept, which can be traced back to works in the early 2000s [1]. Our intention is not to claim novelty at this generic level. Instead, our contribution lies in: **a learning-based, decoupled architecture where integrity verification is governed by a learned, operation-selective fingerprint, while the watermarking scheme acts only as a carrier**. To make this clearer, we summarize the relevant design in the table below.
>
> | Method | Learning-based | Pattern | Adapt to new operations | Robust to benign | Sensitive to tamper |
> |-|-|-|-|-|-|
> | Classical semi-fragile watermarking | × | fixed | hard | ✓ | ✓ |
> | Learning-based semi-fragile watermarking | ✓ | fixed | hard | ✓ | × |
> | Classical watermarking + handcrafted fingerprint | × | fixed | hard | ✓ | ✓ |
> | **SpeeCheck (ours)** | ✓ | flexible | easy | ✓ | ✓ |
>
> **1. Different mechanism: specific watermarking design → watermarking payload**
>
> For semi-fragile watermarking, the authentication behavior is controlled by the **watermarking embedding scheme** itself. The watermark must be placed in signal regions that react differently to benign operations and malicious tampering. For example, to protect image integrity, a common strategy [13] is to embed stable bits in low-frequency DCT coefficients so that they survive JPEG compression, while embedding sensitive bits in high-frequency or edge-related regions that change significantly under local tampering. This creates a rigid **coupling** between the embedding scheme and the target operations. As surveyed in Yu et al. [2], many new semi-fragile schemes are proposed because previous ones can not resist novel attacks. In other words, adapting to new operations often requires designing a new embedding scheme, which makes adaptation hard.
>
> Although there exist learning-based semi-fragile watermarking schemes for images [3,4], it is still difficult to construct a truly semi-fragile audio watermark using current neural audio watermarking designs. Neural audio watermarking systems such as AudioSeal [5] and TimbreWM [6] are optimized to be robust: the watermark is usually embedded in the frequency domain but repeated across time. This repetition is well-suited for robust detection, but it makes it difficult to achieve controlled fragility. For example, deleting a trailing segment of the audio cannot be detected by such schemes.
>
> SpeeCheck uses a different mechanism, as reflected in the last row of the above table. We introduce an explicit decoupled architecture that separates a robust watermark carrier from an operation-selective payload. The watermarking scheme is only a carrier and does not control robustness or fragility. Instead, integrity verification is governed entirely by the embedded payload, namely the learned fingerprint. This fingerprint is trained to be stable under benign operations and sensitive to malicious tampering, without any change to the embedding scheme. In our implementation, we use a state-of-the-art watermarking scheme, AudioSeal [5], purely as a carrier, and it can be replaced or upgraded by any other audio watermarking scheme. Since discriminative power comes from the fingerprint rather than from the embedding scheme, SpeeCheck can easily adapt to new operations by retraining the fingerprint extractor, instead of redesigning the watermarking scheme.

---

> ### Author Response · Authors · 2025-11-21
> **Author Response (2/n)**
>
> **2. Distinction between the learned fingerprint and perceptual hashing**
>
> Although SpeeCheck produces a compact representation, it is different from standard perceptual hashing in at least three aspects.
>
> * First, the **design objective** is different. Classical perceptual hashes for audio or images are typically designed for robust matching in **retrieval or duplicate detection** [7,8]. They are designed to remain stable under a broad set of signal-level changes as long as content identity is preserved. In contrast, the SpeeCheck fingerprint is explicitly trained to be operation-selective: it is robust only to benign operations while remaining sensitive to malicious tampering.
>
> * Second, the **verification mechanism** is different. Traditional perceptual hashing typically relies on storing hashes in an **external database** and performing query-based matching. By design, SpeeCheck embeds the fingerprint directly into the speech via watermarking and recomputes it from the received signal, so verification is self-contained and does not require any external reference database.
>
> * Third, while some works [1] have combined perceptual hashing with watermarking to avoid external storage (row 3 in above table), these systems are still similar to traditional semi-fragile watermarking: the watermark pattern and the used features must be designed carefully for a specific set of benign operations and attacks. Most existing perceptual hashes rely on **handcrafted features** [9,10], which makes it difficult to extend them to new manipulation types. In contrast, SpeeCheck uses a learned multiscale fingerprint trained with contrastive learning to separate benign operations from malicious tampering. This is analogous to the shift from handcrafted spectral features such as MFCCs [11] to learned representations such as wav2vec 2.0 [12] in modern speech processing, which has led to stronger discriminative representations across many tasks.
>
> We appreciate the reviewer’s insightful perspective. We will revise the paper to avoid overstatement and highlight contributions more clearly:
> * a decoupled architecture that separates a robust watermark carrier from a learned fingerprint that controls integrity verification, and
> * a learning-based multiscale fingerprint that enables self-contained, operation-selective speech integrity verification.
>
> ---
> **References:**
>
> [1] Gomez, Emilia, et al., 2002. Mixed watermarking-fingerprinting approach for integrity verification of audio recordings.
>
> [2] Yu, Xiaoyan, et al., 2017. Review on semi-fragile watermarking algorithms for content authentication of digital images.
>
> [3] Zhao, Yuan, et al., 2024. Proactive image manipulation detection via deep semi-fragile watermark.
>
> [4] Neekhara, Paarth, et al., 2024. Facesigns: Semi-fragile watermarks for media authentication.
>
> [5] San Roman, Robin, et al., 2024. Proactive Detection of Voice Cloning with Localized Watermarking.
>
> [6] Liu, Chang, et al., 2024. Detecting Voice Cloning Attacks via Timbre Watermarking.
>
> [7] Microsoft. PhotoDNA. 2015. Available at: https://www.microsoft.com/en-us/photodna.
>
> [8] Facebook. Open-sourcing photo- and video-matching technology to make the Internet safer. 2019. Available at: https://about.fb.com/news/2019/08/open-source-photo-video-matching.
>
> [9] Seo, Jin S., et al., 2006. Audio fingerprinting based on normalized spectral subband moments.
>
> [10] Seo, Jin S., et al., 2005. Audio fingerprinting based on normalized spectral subband centroids.
>
> [11] Davis, Steven, and Paul Mermelstein, 1980. Comparison of parametric representations for monosyllabic word recognition in continuously spoken sentences.
>
> [12] Baevski, Alexei, et al., 2020. wav2vec 2.0: A framework for self-supervised learning of speech representations.
>
> [13] Lin, Ching-Yung, et al., 2001. A robust image authentication method distinguishing JPEG compression from malicious manipulation.

---

> ### Author Response · Authors · 2025-11-21
> **Author Response (3/n)**
>
> > **[Q2] Watermarking should not destroy the perceptual hash. No consideration of this topic is given here.**
>
> Thank you for this important comment.
>
> We fully agree that the watermarking process **must not distort** the acoustic fingerprint it is designed to protect; otherwise the verification procedure would be fundamentally flawed.
>
> This constraint was a key consideration in our design. As stated in the paper **(Lines 314–316)**, we already verified that the acoustic fingerprint computed from the watermarked audio remains consistent with that from the original audio, while the embedded bits can still be reliably recovered.
>
> This is achieved for two reasons:
>
> * The watermarking scheme, AudioSeal, is designed to introduce only minor and imperceptible perturbations to the waveform.
>
> * Our fingerprint generation is trained via contrastive learning to be robust to benign operations such as compression and resampling. The watermarking perturbation behaves similarly to such benign transformations, so the fingerprint is encouraged to remain stable under the presence of the watermark.
>
> We agree that this point is critical and deserves a more explicit quantitative evaluation. Following your suggestion, we have conducted an additional experiment to measure the impact of watermarking on the fingerprint. For each audio sample, we compute the fingerprint from the original audio and from its watermarked version, and then report the average Hamming distance between these two fingerprints.
>
> | Dataset | VoxCeleb | LibriSpeech | Real-world |
> |---------|----------|-------------|------------|
> | Average Hamming Distance | 6.87 | 9.15 | 5.40 |
>
> Given that our fingerprint length is 256 bits and the decision threshold is θ = 42, these distances (e.g., a maximum average of 9.15) are very small and significantly blow the detection threshold. This confirms that watermarking has only a minor effect on the fingerprint and does not compromise the integrity verification logic. We will add this experiment and analysis to the revised paper.
>
> > **[Q3] More broadly, it would be good to discover the reason behind false negatives: is it due to a watermarking decoding failure or due to the feature extraction failure?**
>
> Thank you for this question, which allows us to clarify how different error cases arise in SpeeCheck.
>
> **1. Clarification of terminology**
>
> To avoid confusion, we first restate the label convention in our paper. In our experiments (Tables 1 and 2), we define Benign as the positive class and Tampering as the negative class. Under this convention:
>
> * **False Negative (FN)** means that a benign sample is incorrectly rejected (for example, the overall FNR=1.03% in Table 2).
>
> * **False Positive (FP)** means that a tampered sample is incorrectly accepted (for example, the overall FPR=0.90% in Table 1).
>
> In many tampering detection works, tampering samples are treated as the positive class, so a “false negative” often refers to a missing tampering attack. In our notation, this case corresponds to a False Positive, while False Negatives in our tables correspond to benign speech that is incorrectly rejected. Below, we discuss both error cases explicitly.
>
> **2. False positives (missed tampering)**
>
> These cases are **not caused by watermarking decoding failure**, but by insufficient sensitivity of the fingerprint generation path. SpeeCheck is designed with two decoupled paths:
>
> * Path A (recomputed fingerprint): from the received (possibly tampered) audio, the model recomputes an acoustic fingerprint. This path is designed to be sensitive and should change significantly under malicious edits.
>
> * Path B (embedded fingerprint): from the same audio, the watermark decoder recovers the embedded fingerprint. This path is designed to be robust to benign operations and has very low bit error in our experiments.
>
> A missed tampering (FP) occurs only when these two fingerprints remain too similar, so that their Hamming distance stays below the decision threshold. In practice, this happens when the malicious edit is very subtle and Path A does not reflect a large enough change, while Path B still correctly recovers the original fingerprint. This is happened in our worst-performing category: Substitution (minor) (TNR = 88.40% in Table 10). For all other tampering categories, TNR is close to 100%, which indicates that these missed detections are rare and tied to extremely subtle manipulations.

---

> > ### Comment · Reviewer_JyhN · 2025-11-22
> >
> > > To avoid confusion, we first restate the label convention in our paper.
> >
> > You do not restate, you state. If I remember correctly your paper, these definitions were missing. And as you said, many papers take the opposite definitions.
> >
> > I must read again all the experimental works... very upsetting.

---

> > > ### Author Response · Authors · 2025-11-22
> > > **Follow-up Responses (1/4)**
> > >
> > > > **[Q1] You do not restate, you state. If I remember correctly your paper, these definitions were missing. And as you said, many papers take the opposite definitions. I must read again all the experimental works... very upsetting.**
> > >
> > > We apologize for the confusion and any extra effort this may have caused. However, we would like to clarify that the current submission **already defines** our label convention in several places.
> > >
> > > * In the method section, when we describe contrastive learning, “positive pairs” refer to benign variants and “negative pairs” refer to tampered variants.
> > >
> > > * In Section 4.1 (Evaluation Metrics), we state that “benign operations are treated as positive and malicious ones as negative.”
> > >
> > > * The captions of Tables 1 and 2 also state that benign operations are treated as the positive class and malicious operations as the negative class.
> > >
> > > And we follow this convention throughout the evaluation. We also confirm that this definition has not changed between the paper and the rebuttal; the same convention is used from the beginning.
> > >
> > > The reason for this choice is that we frame the problem as **speech integrity verification** rather than pure **tampering detection**. The system decides whether a given audio sample should be accepted as authentic or rejected as tampered, so in our setting, it is natural to treat benign samples as the positive class and tampered samples as the negative class.
> > >
> > > In the revised version, we highlight this convention more clearly to avoid confusion for readers.

---

> ### Author Response · Authors · 2025-11-21
> **Author Response (4/n)**
>
> **3. False negatives (rejected benign)**
>
> False negatives in our tables correspond to benign speech that is incorrectly classified as tampered. These errors arise from two main reasons:
>
> * For a small subset of benign samples, some allowed operations (e.g., aggressive compression, noise suppression) can induce relatively large acoustic changes. In these cases, the recomputed fingerprint in Path A may drift far from the embedded fingerprint, and the Hamming distance can exceed the threshold, leading to a rejection of a benign sample.
>
> * In a smaller number of cases, the watermarking carrier itself may not be that robust, and the recovered fingerprint from Path B may contain relatively large bit errors. Since SpeeCheck uses AudioSeal as a modular carrier, this component can be replaced by other watermarking scheme in the future if a more robust watermarking is proposed.
>
> We will add this analysis of both false positives (missed tampering) and false negatives (rejected benign) to the Appendix in the revised version. Thank you for this insightful suggestion.
>
> > **[Q4] Localization: This literature not only decides whether the content is authentic but also spots where the manipulation happened within some granularity. There is no localization in this submission.**
>
> Thank you for raising this point. We agree that localization is an important capability for many forensic scenarios. Our current version of SpeeCheck does not provide localization, and we already state this explicitly as a limitation and a direction for future work in the Discussion (Section E).
>
> We respectfully clarify that tampering localization, while valuable for forensics, was **outside the scope of our application scenario**. As defined in Section 2.1, our target use case is to provide general users (for example, journalists, content consumers, and platforms) with a **convenient** and **reliable** tool that returns a clear **utterance-level decision**: “Is this speech authentic or tampered?” In this scenario, the primary requirement is a robust and clear binary judgement for each utterance, rather than fine-grained identification of where exactly the manipulation occurred.
>
> The technical design of SpeeCheck follows this goal. The acoustic fingerprint is learned via contrastive learning as a single utterance-level representation that summarizes the entire speech segment. The training objective optimizes this representation for utterance-level integrity verification under benign operations, instead of producing a time-aligned or region-based map that would be needed for localization.
>
> Within this stated scope, SpeeCheck fulfills its objective by delivering high-accuracy utterance-level integrity verification while remaining robust to common benign processing. We fully agree that extending SpeeCheck with temporal or segment-level resolution would be valuable, especially for forensic applications, and we will consider extending SpeeCheck to include localization in our future research.

---

> ### Author Response · Authors · 2025-11-21
> **Author Response (5/n)**
>
> > **[Q5] Security. The audio watermarking technique AudioSeal is on the shelves, The proposed feature extraction might be as well. Therefore, there is no secret key. An attack can manipulate the content, recalculate the new feature and re-embed it in the audio stream. No consideration about the security in the submission.**
>
> Thank you for raising this important security concern.
>
> The **“replay” attack** you describe is indeed a critical threat model that must be considered. In this current submission, our primary focus is on designing and evaluating a robust, self-contained fingerprinting and verification framework. We did not explicitly highlight security mechanisms against such attacks, but our framework is **compatible with existing security solutions [1-3]**. In particular, it can be strengthened in a straightforward way by introducing **a secret key** into the fingerprint binarization step (Step 4 in Section 3.2).
>
> Concretely, in our current design, the final binary fingerprint is obtained from a continuous feature vector $v \in \mathbb{R}^d$ through a sign operator: $b = \text{Sign}(v)$. To secure this step, we can introduce a secret orthogonal matrix $R \in \mathbb{R}^{d \times d}$ which acts as our private key. The binarization becomes: $b_{\text{secret}} = \text{Sign}(v R)$
>
> This modification provides two benefits:
>
> **1) Attackers cannot forge a valid fingerprint.**
>
> Even if an attacker can recompute a feature vector $v_{\text{tampered}}$ from a manipulated audio, they do not have access to the secret matrix $R$, and therefore **cannot generate the correct secured fingerprint** $b_{\text{secret}}$ that will be accepted by the verifier. The non-linear $\text{Sign}$ function further removes magnitude information and makes it difficult to infer $R$ from observed binary outputs.
>
> **2) Robustness is preserved.**
>
> Since $R$ is orthogonal, it acts as a rotation that preserves distances in the feature space. As a result, this keyed transformation does not change the robustness properties of SpeeCheck: benign operations and malicious tampering still induce similar distance patterns in the transformed space, so the decision rule and performance remain essentially **unchanged**.
>
> This keyed-fingerprint mechanism integrates naturally with our SpeeCheck and provides a clear defense against the replay attack. We will add a dedicated subsection in the revised version to explain this extension and discuss how the secret transform can be managed in practice. In addition, in many realistic deployments, the fingerprint generator and the watermark embedder are not released as public, but are provided as managed services. While we do not rely on obscurity as a formal security assumption, such a deployment further raises the bar for attackers.
>
> We acknowledge that a full formal security analysis is beyond the scope of the current work. In the revised paper, we will
>
> * clarify the replay-style threat model,
>
> * explain how a keyed binarization step can be used to secure SpeeCheck against such attacks, and
>
> * highlight the security extensions as an important direction for future work.
>
> ---
> **References:**
>
> [1] Jin, Andrew Teoh Beng, et al., 2004. Biohashing: two factor authentication featuring fingerprint data and tokenised random number.
>
> [2] Charikar, Moses S, 2002. Similarity estimation techniques from rounding algorithms.
>
> [3] Evennou, Gautier, et al., 2025. Fast, Secure, and High-Capacity Image Watermarking with Autoencoded Text Vectors.

---

> ### Author Response · Authors · 2025-11-21
> **Author Response (6/n)**
>
> > **[Q6] Since the feature extraction is the main contribution, a dedicated benchmark is welcome.**
>
> Thank you for this constructive suggestion. We agree that the fingerprint generation (feature extraction) module is a key part of our contribution, and we are happy to provide a more dedicated analysis.
>
> Our work has two primary contributions:
>
> * a decoupled, self-contained verification framework that separates a robust watermarking carrier from an operation-selective fingerprint payload, and
>
> * a learning-based fingerprint generation module, which uses multiscale feature extraction and binarization to produce discriminative acoustic fingerprints.
>
> The framework itself is important because it separates the roles of the watermarking carrier and the discriminative fingerprint. This makes SpeeCheck extensible and adaptable: a stronger watermarking scheme can be plugged in without changing the fingerprint generator, and the fingerprint generator can be fine-tuned to new benign operations or new attacks without redesigning the watermarking algorithm.
>
> Regarding the feature extraction module, the current submission already includes several analyses. Figure 4(a) shows that the learned multiscale features have high cosine similarity for benign pairs (near 1.0) and low similarity for malicious or cross pairs. Figure 4(b) shows that this separation is preserved after binarization, with a large margin in Hamming distance between benign and malicious pairs. Figure 5 (and Figure 8) provides t-SNE visualizations showing that, after training, benign samples cluster tightly with their anchors, while malicious samples are pushed away. Table 15 presents an ablation study where removing the multiscale module degrades overall performance.
>
> To further evaluate the importance of the feature extraction part, we ran an additional experiment that removes the multiscale module and uses only the BiLSTM output as the fingerprint feature. The training protocol, datasets, and evaluation metrics remain exactly the same as in the main experiments; only the multiscale branch is disabled.
>
> | Operation         | TPR    | FPR   | AUC    | EER  |
> |-------------------|--------|-------|--------|------|
> | Resampling        | 99.40  | 11.00 | 99.52  | 3.20 |
> | Compression       | 79.60  | 13.40 | 90.41  | 14.40 |
> | Reencoding        | 100.00 | 10.20 | 99.85  | 0.90 |
> | Noise suppression | 100.00 | 11.00 | 99.84  | 0.70 |
> | **Overall**           | 94.75  | 11.40 | 97.41  | 4.80 |
>
> | Operation               | TNR    | FNR   | AUC    | EER   |
> |-------------------------|--------|-------|--------|-------|
> | Deletion (minor)        | 83.60  | 6.00  | 97.53  | 9.30  |
> | Deletion (moderate)     | 97.40  | 5.60  | 99.22  | 4.00  |
> | Deletion (severe)       | 99.40  | 4.60  | 99.68  | 1.60  |
> | Splicing (minor)        | 96.00  | 4.00  | 99.16  | 4.00  |
> | Splicing (moderate)     | 98.60  | 4.80  | 99.60  | 2.70  |
> | Splicing (severe)       | 99.60  | 6.00  | 99.83  | 2.10  |
> | Silencing (minor)       | 79.60  | 5.20  | 96.37  | 13.70 |
> | Silencing (moderate)    | 82.00  | 5.20  | 97.00  | 11.60 |
> | Silencing (severe)      | 85.80  | 4.00  | 97.80  | 9.60  |
> | Substitution (minor)    | 64.60  | 5.40  | 92.55  | 20.80 |
> | Substitution (moderate) | 70.60  | 5.60  | 94.45  | 18.10 |
> | Substitution (severe)   | 81.00  | 4.20  | 96.74  | 13.00 |
> | Reordering              | 93.20  | 6.60  | 97.85  | 6.70  |
> | Text-to-speech          | 100.00 | 0.00  | 100.00 | 0.00  |
> | Voice Conversion        | 100.00 | 4.80  | 99.75  | 2.40  |
> | **Overall**                 | 87.55  | 4.74  | 98.09  | 7.97  |
>
> For benign operations, without the multiscale module, the overall performance drops to TPR = 94.75%, FPR = 11.40%, AUC = 97.41%, and EER = 4.80%. For tampering attacks, the impact is more severe. The overall performance becomes TNR = 87.55%, FNR = 4.74%, AUC = 98.09%, and EER = 7.97%. The degradation is especially clear for **subtle tampering attacks** such as Substitution (minor), where the **EER** increases from **6.10%** (with multiscale features) to **20.80%** (without the multiscale module), and **TNR** drops to **64.60%**.
>
> These new results provide a more comprehensive evaluation of the feature extraction module and support our claim that multiscale design is essential for maintaining sensitivity to subtle tampering while preserving robustness to benign operations. We will add this experiment to the appendix in the revised version.

---

> > ### Comment · Reviewer_JyhN · 2025-11-22
> >
> > What I mean by a dedicated benchmark is a comparison with many audio fingerprinting schemes.
> >
> > The only comparison in the paper is for deepfake detection, and with two old techniques (not from the latest ASVSpoof edition).

---

> > > ### Author Response · Authors · 2025-11-23
> > > **Follow-up Responses (2/4)**
> > >
> > > > **[Q2] What I mean by a dedicated benchmark is a comparison with many audio fingerprinting schemes. The only comparison in the paper is for deepfake detection, and with two old techniques (not from the latest ASVSpoof edition).**
> > >
> > > Thank you very much for the clarification.
> > >
> > > We agree that the fingerprint generation module is a key part of our contribution and that a more dedicated comparison is important. Existing audio fingerprinting and perceptual hashing methods are primarily designed for **content retrieval**, where they are tuned to remain stable under very severe distortions, which makes them hard to use directly for integrity verification. After a broad survey of works that explicitly address integrity verification, we only found **non-learning-based** schemes [1-4] that rely on handcrafted features, and they do not open-source their implementations. Nevertheless, we agree that a fingerprint-level benchmark is important. In the revised version, we plan to reproduce these audio fingerprinting methods based on their papers and evaluate them under our benign and malicious operation sets.
> > >
> > > For audio deepfake detection, we originally chose RawNet2 and AASIST because they are widely used baselines in the ASVspoof literature. Following your suggestion, we have identified **RawGAT-ST** [5] as a more recent method that extends RawNet2. In the revised version, we will evaluate the performance of RawGAT-ST and replace RawNet2 with RawGAT-ST in the main comparison.
> > >
> > > We are currently running these experiments and will update the revised manuscript once the new results are available.
> > >
> > > ---
> > > **References:**
> > >
> > > [1] Renza, Diego, et al., 2019. Robust speech hashing for digital audio forensics.
> > >
> > > [2] Shi, Canghong, et al., 2020. A novel integrity authentication algorithm based on perceptual speech hash and learned dictionaries.
> > >
> > > [3] Zhang, Qiu-yu, et al., 2018. An efficient perceptual hashing based on improved spectral entropy for speech authentication.
> > >
> > > [4] Zhang, Qiu-yu, et al., 2021. An encrypted speech authentication and tampering recovery method based on perceptual hashing.
> > >
> > > [5] Tak, Hemlata, et al., 2021. End-to-end spectro-temporal graph attention networks for speaker verification anti-spoofing and speech deepfake detection.

---

> > > ### Author Response · Authors · 2025-11-24
> > > **Follow-up Response (5)**
> > >
> > > > **[Q2] What I mean by a dedicated benchmark is a comparison with many audio fingerprinting schemes. The only comparison in the paper is for deepfake detection, and with two old techniques (not from the latest ASVSpoof edition).**
> > >
> > > Thank you for your clarification and patience.
> > >
> > > We have completed the **audio fingerprinting benchmark** that you suggested. In addition to SpeeCheck’s learned fingerprint, we implement and evaluate five representative audio fingerprinting or perceptual hashing schemes in the integrity verification setting. The table below reports their performance on VoxCeleb and LibriSpeech under the same benign and malicious operation sets as SpeeCheck:
> > >
> > >
> > > | Method | TPR (Vox) | FPR (Vox) | AUC (Vox) | EER (Vox) | TPR (Libri) | FPR (Libri) | AUC (Libri) | EER (Libri) |
> > > |-------|-----------|-----------|-----------|-----------|-------------|-------------|-------------|-------------|
> > > | Chromaprint [1]| 93.06 | 16.30 | 92.34 | 12.58 | 81.09 | 13.43 | 85.38 | 16.68 |
> > > | Renza et al. [2] | 82.40 | 3.35 | 93.85 | 12.37 | 74.90 | 2.05 | 89.67 | 18.53 |
> > > | Shi et al. [3] | 60.20 | 8.30 | 79.88 | 26.12 | 59.75 | 8.55 | 80.93 | 26.95 |
> > > | Zhang et al. [4] | 85.65 | 17.40 | 93.07 | 15.85 | 79.38 | 12.73 | 91.96 | 16.70 |
> > > | Zhang et al. [5] | 92.05 | 6.14 | 97.43 | 7.24 | 90.59 | 8.65 | 96.94 | 9.05 |
> > > | **SpeeCheck (only fingerprint)** | 99.32 | 0.52 | 99.98 | 0.55 | 98.24 | 2.11 | 99.67 | 1.96 |
> > >
> > > These handcrafted schemes struggle to balance the acceptance of bening samples and the rejection of malicious ones, which is not suitable for integrity verification. In addition, their handcrafted features are tightly coupled to the specific operations considered in the original papers, and adapting them to new types of benign or malicious operations would require manual redesign. In contrast, SpeeCheck’s fingerprint exhibit clear capability for distinguishing malicious and benign, which is exactly required for integrity verification. We have added this benchmark and the corresponding analysis to the main paper (Table 5).
> > >
> > > Regarding the concern about **outdated deepfake detection baselines**, we have also updated the deepfake detection comparison. As shown in the table below, we adopt Nes2Net [6] and SSL-AntiSpoofing [7], two recent and strong deepfake detectors that build on self-supervised representations and show competitive performance on ASVspoof benchmarks. Both methods achieve strong performance when a large portion of the utterance is replaced by deepfake content, but their performance drops noticeably at low substitution ratios, which indicates limited sensitivity to subtle deepfake edits. In contrast, SpeeCheck consistently maintains strong detection performance across all substitution levels. We have added the new results (Table 4) and the updated discussion into the main paper.
> > >
> > > The revised paper has been uploaded. We hope our response can address your concerns. Once again, thank you for your efforts in helping improve our work.
> > >
> > > | Deepfake Ratio | Nes2Net TPR | Nes2Net FPR | Nes2Net AUC | Nes2Net EER | SSL-AntiSpoofing TPR | SSL-AntiSpoofing FPR | SSL-AntiSpoofing AUC | SSL-AntiSpoofing EER | SpeeCheck TPR | SpeeCheck FPR | SpeeCheck AUC | SpeeCheck EER |
> > > |----------------|------------:|------------:|------------:|------------:|----------------------:|----------------------:|----------------------:|----------------------:|--------------:|--------------:|--------------:|--------------:|
> > > | Substitute 10% | 54.27 | 30.08 | 65.60 | 39.84 | 57.11 | 19.52 | 72.58 | 33.00 | 82.97 | 0.40 | 99.57 | 3.21 |
> > > | Substitute 25% | 88.01 | 20.12 | 90.54 | 16.46 | 71.54 | 19.72 | 81.10 | 25.96 | 98.20 | 0.40 | 99.95 | 0.40 |
> > > | Substitute 50% | 98.78 | 2.03  | 99.89 | 1.83  | 79.27 | 12.27 | 89.29 | 17.71 | 100.00 | 0.40 | 100.00 | 0.30 |
> > > | Substitute 75% | 100.00 | 0.00 | 100.00 | 0.00 | 93.50 | 10.26 | 97.25 | 9.05  | 100.00 | 0.40 | 100.00 | 0.00 |
> > > | Substitute 90% | 100.00 | 0.00 | 100.00 | 0.00 | 95.12 | 3.82  | 98.97 | 4.63  | 100.00 | 0.40 | 100.00 | 0.00 |
> > >
> > > ---
> > > **References:**
> > >
> > > [1] Lukas Lalinsky. Chromaprint. 2010. Available at: https://acoustid.org/chromaprint.
> > >
> > > [2] Renza, Diego, et al., 2019.  Robust speech hashing for digital audio forensics.
> > >
> > > [3] Shi, Canghong, et al., 2020. A novel integrity authentication algorithm based on perceptual speech hash and learned dictionaries.
> > >
> > > [4] Zhang, Qiu-yu, et al., 2018.  An efficient perceptual hashing based on improved spectral entropy for speech authentication.
> > >
> > > [5] Zhang, Qiu-yu, et al., 2021. An encrypted speech authentication and tampering recovery method based on perceptual hashing.
> > >
> > > [6] Liu, Tianchi, et al., 2025. Nes2net: A lightweight nested architecture for foundation model driven speech anti-spoofing.
> > >
> > > [7] Tak, Hemlata, et al., 2022. Automatic speaker verification spoofing and deepfake detection using wav2vec 2.0 and data augmentation.

---

> ### Author Response · Authors · 2025-11-21
> **Author Response (7/n)**
>
> > **[Q7] As for authentication, AudioSeal can hide a 16-bit message, but its decoder also gives a detection score per sample. The absence of watermark signifies the manipulation. A comparison with AudioSeal is definitely missing. (However, I assume here that one can learn a private AudioSeal)**
>
> Thank you for this thoughtful question.
>
> We agree that AudioSeal can embed a 16-bit message as well as output a detection score (zero-bit) per sample. However, we respectfully disagree that AudioSeal’s detection score is suitable for integrity verification, due to a fundamental conflict in design objectives, and we support this with a dedicated experiment.
>
> Conceptually, AudioSeal is a **robust watermarking** method. Its objective is to maintain a high detection score as long as the embedded watermark survives, even under strong signal distortions. **This is a suitable design for copyright protection, but it conflicts with the needs of integrity verification**. For example, if an attacker deletes a few words from a speech, the remaining audio still contains valid AudioSeal watermarks. The detection score stays high, even though the semantic content has changed. Likewise, if segments are reordered, the watermark attached to each segment simply follows the segment, and the global detection score remains high. In both cases, the watermark confirms that the audio contains the original owner’s watermark, but it does not say whether the content has been tampered.
>
> We conducted an additional experiment using AudioSeal’s detection score as the verification metric. Swept the decision threshold to find a very strict setting that improves sensitivity to attacks. The best threshold we found for this purpose is **θ=0.9997**. Even under this extreme setting, the results are:
>
> | Operation         | TPR    | FPR   | AUC    | EER   |
> |-------------------|--------|-------|--------|-------|
> | Resampling        | 100.00 | 30.93 | 84.12  | 19.59 |
> | Compression       | 100.00 | 19.59 | 88.27  | 16.49 |
> | Reencoding        | 100.00 | 25.77 | 87.00  | 23.20 |
> | Noise suppression | 100.00 | 26.80 | 85.54  | 24.74 |
> | **Overall**           | 100.00 | 24.22 | 86.98  | 26.55 |
>
> | Operation               | TNR    | FNR   | AUC    | EER   |
> |-------------------------|--------|-------|--------|-------|
> | Deletion (minor)        | 0.00   | 0.00  | 52.29  | 47.94 |
> | Deletion (moderate)     | 0.00   | 0.00  | 49.93  | 50.00 |
> | Deletion (severe)       | 0.00   | 0.00  | 45.45  | 54.12 |
> | Splicing (minor)        | 100.00 | 0.00  | 100.00 | 0.00  |
> | Splicing (moderate)     | 100.00 | 0.00  | 100.00 | 0.00  |
> | Splicing (severe)       | 100.00 | 0.00  | 100.00 | 0.00  |
> | Silencing (minor)       | 100.00 | 0.00  | 100.00 | 0.00  |
> | Silencing (moderate)    | 98.97  | 0.00  | 99.32  | 0.52  |
> | Silencing (severe)      | 100.00 | 0.00  | 100.00 | 0.00  |
> | Substitution (minor)    | 100.00 | 0.00  | 100.00 | 0.00  |
> | Substitution (moderate) | 100.00 | 0.00  | 100.00 | 0.00  |
> | Substitution (severe)   | 100.00 | 0.00  | 100.00 | 0.00  |
> | Reordering              | 0.00   | 0.00  | 50.42  | 50.00 |
> | Text-to-speech          | 100.00 | 0.00  | 100.00 | 0.00  |
> | Voice Conversion        | 100.00 | 0.00  | 100.00 | 0.00  |
> | **Overall**                 | 73.26  | 0.00  | 86.49  | 13.51 |
>
> These results show that, even with this strict threshold, AudioSeal’s detection score alone does not provide a reliable integrity verification mechanism: it cannot detect common tampering patterns such as deletion and reordering, and it requires an impractically strict threshold that rejects many benign samples.
>
> > **[Q8] The motivation summarized in Figure 2 is naive. No professional would ever consider SHA256 hash as a suitable feature representation for multimedia content.**
>
> Thank you for this comment. We agree with the reviewer that SHA256 is not a suitable feature representation for multimedia content, and we do not use it as a feature in our motivation part.
>
> Cryptographic hash functions such as SHA256 or MD5 are widely used as the industry standard for digital data integrity verification. Our intention in Figure 2 is to use SHA256 as a simple and familiar example of such conventional integrity verification, not as a proposal for a multimedia feature extractor.
>
> The purpose of Figure 2 is to highlight two fundamental limitations of conventional hash-based integrity verification when applied to speech:
> * they are overly sensitive, and
> * they need an external reference for verification.
>
> These limitations directly motivate the design requirements for SpeeCheck: build a system that is robust to benign operations but sensitive to tampering, and that supports self-contained verification without external hash storage.

---

> > ### Comment · Reviewer_JyhN · 2025-11-22
> >
> > > Our intention in Figure 2 is to use SHA256 as a simple and familiar example
> >
> > I understand. What I am saying is that this illustration is so simple and naive that a reader expert in the field may stop and disregard your paper. This example is more dissuasive than helpful.

---

> > > ### Author Response · Authors · 2025-11-23
> > > **Follow-up Responses (3/4)**
> > >
> > > > **[Q3] I understand. What I am saying is that this illustration is so simple and naive that a reader expert in the field may stop and disregard your paper. This example is more dissuasive than helpful.**
> > >
> > > Thank you for the clarification.
> > >
> > > We agree that the SHA256-based illustration is very simple and does not provide additional technical insight for expert readers. In the revised version, we therefore **remove Figure 2 (b)** (The distribution of SHA256 Hamming distance to the original audio under different modifications). Instead, Section 2.4 now presents a concise textual discussion that summarizes **hash-based integrity verification**, covering both cryptographic hashes and perceptual hashes, and directly explains why these methods are either overly sensitive or difficult to adapt to new operations. We believe this revision keeps the motivation focused and highlights the limitations that SpeeCheck is designed to address.

---

> ### Author Response · Authors · 2025-11-21
> **Author Response (8/n)**
>
> > **[Q9] I have some concerns about the evaluation metric (paragraph Line 337). Note that the threshold is fixed.
> How can you compute an AUC if the threshold is fixed? Anyway, AUC is a really bad metric for applications requiring small FPR.
> How can you compute an EER if the threshold is fixed? Why is the FPR varying in Table 1? Table 1 deals only with positive (ie. benign) content. Therefore, there is no way to measure a FPR as this requires negative (ie. manipulated) content. Moreover, the fact that the FPR varies from one benign operation to another can not be correct. The same happens with Table 2 where FNR evaluations are wrong. Also, note that, in principle, FNR + TPR = 100 and FPR + TNR = 100, which is not the case here. Conclusion: Table 1 should report only TPR and Table 2 only FPR.**
>
> Thank you for raising these questions about the evaluation protocol and metrics. They are helpful for clarifying our setup.
>
> **1) AUC and EER versus a fixed threshold**
>
> We agree that AUC and EER are **threshold-independent** metrics, while TPR/FPR/TNR/FNR depend on a specific decision threshold. In our experiments, we sweep the decision threshold over the entire range to obtain the ROC curve and compute AUC. Similarly, using the same sweep, we compute EER as the point where FPR and FNR are equal. After that, we choose a single operating threshold θ on the validation set. Then this fixed θ is used on the test set to report TPR/FPR/TNR/FNR in Tables 1 and 2. We agree that AUC can be less informative for very small-FPR regimes, and this is why we also report explicit FPR and TNR at the chosen θ.
>
> **2) Construction of Tables 1 and 2 and the meaning of FPR/FNR**
>
> We also respectfully clarify a misunderstanding regarding the experimental setup for Tables 1 and 2. Contrary to the statement “Table 1 deals only with positive (i.e., benign) content,” each row in Table 1 is evaluated on a balanced binary classification task that **contains both benign and tampered samples**, as explicitly stated in **Lines 371-374** (Section 4.2).
>
> For a given benign operation $O$ (e.g., resampling),
>
> * Positives: audio samples with operation $O$ applied (benign).
>
> * Negatives: an equal number of tampered samples, randomly drawn from the pool of malicious operations.
>
> We repeat this sampling to build a balanced test set for each row (operation). Then:
>
> * **TPR in Table 1** = fraction of those benign $O$-samples that are accepted.
> * **FPR in Table 1** = fraction of the sampled malicious attacks that are incorrectly accepted under the same threshold.
>
> Because the negative subset is resampled per operation, the distribution of attacks differs from row to row, and therefore the observed FPR can vary across different benign operations. The same logic applies to FNR in Table 2.
>
> **3) Consistency of TPR/FNR and TNR/FPR**
>
> We agree with the principle that FNR + TPR = 100 and FPR + TNR = 100. Our results are consistent with this principle. The variations are solely due to the randomized sampling described above.
>
> We appreciate the reviewer’s detailed comments and will make the evaluation protocol and metric definitions in Section 4.2 more explicit in the revised version.

---

> ### Author Response · Authors · 2025-11-21
> **Author Response (9/n)**
>
> > **[Q10] Since most of the operations deal with the semantics, why not hiding the transcript Speech-to-Text inside the audio stream with the help of LLM zip. Like this paper did for images: FAST, SECURE, AND HIGH-CAPACITY IMAGE WATERMARKING WITH AUTOENCODED TEXT VECTORS, Evennou et al.**
>
> Thank you for this interesting idea (Evennou et al.).
>
> We agree that embedding autoencoded text vectors or compressed transcripts is very effective for **semantic communication**. However, this design is fundamentally **unsuitable for speech integrity verification**, for two main reasons.
>
> **1) Blindness to speaker identity attacks.**
>
> As defined in Section 2.1 **(Line 127-128)**, SpeeCheck is designed to detect two types of attacks:
>
> * **semantic tampering**: changing what was said.
>
> * **speaker identity tampering**: changing who said it (e.g., voice conversion).
>
> A transcript-based watermark fails completely for **attack#2**. Modern Speech-to-Text (STT) models are intentionally speaker invariant: they remove timbre, prosody, and other paralinguistic features to focus on text. As a result, a voice conversion attack that changes the speaker identity but preserves the transcript would still produce the same STT output. A system that hides only the transcript inside the audio would therefore accept such attacks, because the embedded and recomputed text vectors would match, even though the speaker identity has been altered. This violates our threat model, where identity tampering must be treated as a manipulation.
>
> In contrast, SpeeCheck’s fingerprint is derived directly from acoustic representations and is trained to be sensitive to both semantic and speaker changes, so that voice conversion and other identity attacks are detected.
>
> **2) Efficiency and capacity.**
>
> LatentSeal (Evennou et al.) approach requires a large captioning model/STT and a text autoencoder to map the text into a continuous latent vector. To apply this idea to speech, one would need to run STT, text encoding, and then watermark embedding and decoding, and repeat this at verification time. This adds significant computational and system complexity.
>
> SpeeCheck instead learns a compact 256-bit fingerprint directly from frame-level acoustic features. This fingerprint captures both semantic content and speaker identity, without requiring text decoding or re-encoding.
>
> We agree that embedding a transcript can be useful for semantic communication or caption verification. However, for the integrity verification, which must consider both semantic tampering and speaker identity tampering, a transcript-based watermark is not sufficient.

---

> ### Author Response · Authors · 2025-11-21
> **Author Response (10/10)**
>
> > **[Q11] Why the TNR is lower than 100 for the voice conversion operation? As for Text-to-Speech. Isn't it a new audio stream not-watermarked? In the same way, Table 4 is obvious: deepfakes are not watermarked.**
>
> Thank you for this insightful question.
>
> **1. Why is TNR for voice conversion slightly below 100%?**
>
> Voice conversion (VC) is a type of tampering attack that changes who is speaking while preserving what is said. In our pipeline, VC is applied to watermarked audio: we first embed the fingerprint via AudioSeal, and then perform VC on the watermarked audio. This is consistent with our threat model, where authentic public speeches are watermarked at the source and later may be modified by attackers.
>
> Under such VC attacks, the acoustic characteristics is altered. In most samples, the recomputed fingerprint (from the tampered audio) differs strongly from the embedded fingerprint recovered from the watermark decoder, leading to a high TNR (e.g., 99.4%). The small fraction of errors (on the order of 0.6%) corresponds to rare cases where:
>
> * the VC model does not fully change the speaker characteristics or still preserves some acoustic traits of the original speaker; and/or
>
> * the tampered signal remains close to the original in the feature space, such that the recomputed fingerprint is still similar to the embedded one.
>
> **2. TTS / deepfakes and “obvious” detection by watermark absence**
>
> Regarding Text-to-Speech (TTS) and other deepfake audio, we agree with the reviewer that these signals are typically generated as new audio streams without the embedded watermark (calculated fingerprint). In our framework, this is not a weakness but a desired property: any audio that does not contain a valid fingerprint should be treated as untrusted.
>
> During verification, SpeeCheck checks both paths:
>
> * The watermarking path attempts to recover an embedded fingerprint. If the audio is not watermarked (as with a deepfake generated from scratch), this recovery fails or yields a random pattern.
>
> * The feature-extraction path recomputes a fingerprint from the received audio. The Hamming distance between the recomputed fingerprint and the (missing or invalid) embedded fingerprint will be large, and the sample is rejected.
>
> Thus, TTS and other deepfake audios are reliably flagged as tampered, which is exactly what we want in our application scenario.
> We acknowledge that, “deepfakes are not watermarked” may sound obvious once the framework is understood. However, we believe it is still important to report empirical results in Table 2 and Table 4:
>
> * It confirms that the system’s real performance with TTS and VC attacks, not just idealized assumptions.
> * It allows a direct comparison to passive deepfake detectors (i.e., RawNet2 and AASIST), which operate under a different principle.
>
> > **[Q12] Embedding rate. The 256-dimension feature is embedded into 16 segments. How many seconds does this represent? What happen next if the audio stream is longer? Do you extract and embed another feature or is it always the same "global" feature which is embedded repeatedly?**
>
> Thank you for this valuable question.
>
> In our current design, the segment length is not fixed but derived from the utterance duration. SpeeCheck divides the entire input audio of duration $T$ into 16 non-overlapping segments. Therefore, the duration of each segment is $T/16$ seconds (as described in Section 3.2, Step 5). For example, a 20-second clip yields segments of 1.25 seconds, while a 10-minute recording yields segments of 37.5 seconds.
>
> SpeeCheck extracts one global 256-bit fingerprint that summarizes the entire utterance. This fingerprint is divided into sixteen 16-bit chunks, with one chunk embedded into each of the 16 temporal segments using AudioSeal. In other words, we do not repeatedly embed the same global fingerprint over time; instead, we distribute different portions of a single global fingerprint across the full duration of the audio. The effective payload is therefore 256 bits per utterance, and the embedding rate in bits per second is $256/T$, which decreases as the utterance becomes longer.
>
> As the reviewer correctly implies, stretching a single global fingerprint over a very long duration can reduce temporal resolution and affect its sensitivity to local tampering. We observe this effect in our long-duration experiments summarized in Appendix C.7 (Table 11), where SpeeCheck still generalizes reasonably to longer clips but performance gradually degrades as the duration increases.
>
> In a practical deployment for long-form content (e.g., long interviews or podcasts), a natural extension is to process the stream in fixed-length chunks (for example, 20-second windows), compute a separate 256-bit fingerprint for each chunk, and embed these fingerprints in their corresponding segments. This ensures fine-grained tampering detection for long recordings.

---

> > ### Comment · Reviewer_JyhN · 2025-11-22
> >
> > I thank the authors for their very professional rebuttal. It contains many pieces of information that were missing or that I have missed for my first reading.
> >
> > My question now is how the author will integrate all these pieces of information into a new version of the paper.
> > Please, provide a summary of all the modifications that are planned/committed to be implemented.

---

> > > ### Author Response · Authors · 2025-11-23
> > > **Follow-up Responses (4/4)**
> > >
> > > > **[Q4] I thank the authors for their very professional rebuttal. It contains many pieces of information that were missing or that I have missed for my first reading. My question now is how the author will integrate all these pieces of information into a new version of the paper. Please, provide a summary of all the modifications that are planned/committed to be implemented.**
> > >
> > >
> > > Thank you very much for helping us improve our work. We have integrated these points into a revised version, which has been uploaded. We provide a **global response** that summarizes all implemented changes.

---

> > > ### Author Response · Authors · 2025-11-24
> > > **Summary of modifications**
> > >
> > > Thank you for your valuable comments and suggestions. Below, we summarize your main concerns and the corresponding changes in the revised paper. We hope our response can address your concerns.
> > >
> > > ---
> > > > **[Q1] Challenge the contribution and novelty.**
> > >
> > > Section 1: We expanded the related work and restated our contributions more precisely. Appendix A: We added a broader survey of related integrity verification methods.
> > >
> > > > **[Q2] Watermarking should not destroy fingerprint.**
> > >
> > > Appendix C.12: We added empirical results supporting that watermarking does not affect the fingerprint.
> > >
> > > > **[Q3] Suggest analysing false negatives.**
> > >
> > > Appendix C.13: We added a detailed analysis of FN and FP.
> > >
> > > > **[Q4] Lack of tampering localization.**
> > >
> > > Appendix E: We explained and pointed out it as future work.
> > >
> > > > **[Q5] Not considering security.**
> > >
> > > Appendix D: We analysed adversarial attacks and replay attacks, and added a security extension.
> > >
> > > > **[Q6] No benchmarking of fingerprinting.**
> > >
> > > Section 4.2, Table 5: We added an audio fingerprinting benchmark experiment.
> > >
> > > > **[Q7] Compared deepfake methods are old.**
> > >
> > > Section 4.2, Table 4: We replaced the baselines with more recent deepfake detection methods and reevaluated.
> > >
> > > > **[Q8] SHA256 motivation is native.**
> > >
> > > Section 2.4: We removed the SHA256 toy example and instead analyzed hash-based integrity verification (cryptographic and perceptual hashes).
> > >
> > > > **[Q9] Evaluation metric confusion.**
> > >
> > > Section 4: We clarified the metric definitions and evaluation protocol.
> > >
> > > > **[Q10] Embedding rate.**
> > >
> > > Appendix C.7: We extended the segmentation length experiment.
> > >
> > > ---
> > >
> > > Once again, thank you for your efforts in helping improve our work.

---

> > > > ### Author Response · Authors · 2025-11-27
> > > > **Looking forward to your feedback!**
> > > >
> > > > Dear Reviewer JyhN,
> > > >
> > > > Thank you once again for your valuable feedback. We have conducted additional experiments and made revisions to the paper based on your suggestions. As the discussion phase is nearing its conclusion, we would like to know if our responses have addressed your concerns. We look forward to hearing from you.
> > > >
> > > > Best, Authors

---

### Official Review · Reviewer_os3q · 2025-10-27

**Soundness:** 3
**Presentation:** 4
**Contribution:** 4
**Rating:** 8
**Confidence:** 3

**Summary:**

The paper proposes SpeeCheck, a proactive speech integrity verification design, which is sensitive to tampering attacks, robust to benign operations, and self-contained. SpeeCheck is based on several delicate design including multi-scale feature extraction + contrastive learning, and the key observation that the fingerprint of a watermarked audio is almost the same as the fingerprint of the original audio.

**Strengths:**

- The paper accurately points out the insufficiency of the current detection methods.
- The paper proposes a solid solution. The choice of multi-scale feature extraction + contrastive learning is inspiring and works impressively well according to the reported experimental results.
- The observation that the fingerprint of a watermarked audio is almost the same as the fingerprint of the original audio is the key to achieve self-containing.
- The experimental results show that the separation performance between benign operation and malicious operation are excellent.

**Weaknesses:**

It'd be interesting to test the framework's robustness under strong targeted attacks like the ones in [1].

[1] "Can DeepFake Speech be Reliably Detected?." Liu, Hongbin, Youzheng Chen, Arun Narayanan, Athula Balachandran, Pedro J. Moreno, and Lun Wang.

**Questions:**

The TPR and TNR reported in Table 1 and 2, although super close to 100%, are still not 100%. What should we do in these failure cases?

---

> ### Author Response · Authors · 2025-11-21
> **Author Response (1/n)**
>
> Thank you for providing your valuable and constructive feedback! Please find below the responses to each comment.
>
> ---
>
> > **[Q1] It'd be interesting to test the framework's robustness under strong targeted attacks like the ones in [1].**
> > [1] "Can DeepFake Speech be Reliably Detected?." Liu, Hongbin, Youzheng Chen, Arun Narayanan, Athula Balachandran, Pedro J. Moreno, and Lun Wang.
>
> Thank you for this insightful suggestion. We agree that strong targeted adversarial attacks are highly relevant for evaluating robustness, and we will add a dedicated discussion of Liu et al. [1] in the revised version.
>
> The adversarial attacks in Liu et al. (e.g., PGD, I-FGSM, SimBA) are explicitly designed to fool passive deepfake detectors such as RawNet2 and AASIST. These detectors are trained as binary classifiers that separate “Real” and “Fake” based on synthesis artifacts. The adversarial attacks work by optimizing additive, imperceptible perturbations so that a given fake sample crosses the classifier’s decision boundary and is predicted as “Real.”
>
> SpeeCheck follows a fundamentally different verification principle. It does not decide authenticity by classifying a sample as real or fake based on artifacts. Instead, SpeeCheck checks the consistency between **(i) a fingerprint that has been embedded into the audio via watermarking** and **(ii) a fingerprint recomputed from the received audio**. Deepfake audio generated from scratch does not contain a valid embedded fingerprint that matches this verification protocol. Under the threat model in [1], an attacker may add adversarial noise to a deepfake, but this operation cannot by itself create a valid 256-bit fingerprint that is consistent with the internal SpeeCheck pipeline. Therefore, the adversarially perturbed deepfake still fails the fingerprint–watermark consistency check and is rejected.
>
> A different question is how SpeeCheck behaves if adversarial perturbations are added to watermarked benign audio. In this case, the attacks in [1] would amount to adding small perturbations that aim to fool a passive detector, but do not change semantic content or speaker identity. For SpeeCheck, such small additive noise behaves similarly to a benign transformation (e.g., compression). Therefore, unless the perturbation is strong enough to alter semantic content or speaker identity, SpeeCheck should continue to accept the audio as benign. If the perturbation becomes strong enough to change semantics or speaker identity, then it falls into our tampering threat model, and SpeeCheck is expected to reject it.
>
> We believe that explaining why adversarial attack is inherently resisted by our SpeeCheck will help clarify the advantage of a proactive design. We will add this discussion to the revised manuscript.
>
> ---
> **References:**
>
> [1] Liu, Hongbin, et al. "Can DeepFake Speech be Reliably Detected?." arXiv preprint arXiv:2410.06572 (2024).

---

> ### Author Response · Authors · 2025-11-21
> **Author Response (2/2)**
>
> > **[Q2] The TPR and TNR reported in Table 1 and 2, although super close to 100%, are still not 100%. What should we do in these failure cases?**
>
> Thank you for this practical and insightful question.
>
> We acknowledge that achieving both 100% TPR and 100% TNR at a single operating point is theoretically challenging. There is an inherent trade-off between robustness (tolerating benign distortions) and sensitivity (detecting malicious tampering). In our current setting, SpeeCheck achieves **TPR = 98.70%** and **TNR = 99.14%**, so the remaining failure cases are already on the order of about 1%, but they still need to be handled in real deployment. In practice, these rare cases can be managed as follows.
>
> **First**, the decision threshold can be tuned for the target application. SpeeCheck supports flexible operating points.
> * For high-security scenarios such as legal evidence verification or news media authentication, the decision threshold θ = 42 can be lowered to further reduce false acceptances and increase TNR. This will increase false rejections of benign audio, but reduce the risk that a tampered sample is incorrectly accepted.
> * For more user-oriented scenarios such as social media, the current threshold offers a more balanced trade-off between usability and security.
>
> **Second**, borderline cases can be handled through **hierarchical verification**. Instead of using a single hard threshold θ, the system can define a narrow uncertainty band around this value.
> * Samples with Hamming distance well below the band are accepted.
> * Samples well above the band are rejected.
> * Samples that fall inside the band are flagged as uncertain and sent to secondary checks, such as human review or external forensic tools, depending on the deployment.
>
> Although perfect TPR and TNR are not realistic, tuning the threshold for specific security requirements and introducing an explicit uncertainty region allows practical systems to handle the small fraction of edge cases in a controlled and predictable way. We will include this clarification in the revised version.

---

> > ### Author Response · Authors · 2025-11-26
> > **Looking forward to your feedback!**
> >
> > Dear Reviewer os3q,
> >
> > Thank you once again for your valuable feedback.
> >
> > We have made revisions to the paper based on your suggestions. We hope our response can address your concerns. Please feel free to reach out if you require any further clarification. And thank you again for improving our work!
> >
> > Best, Authors

---

> > > ### Comment · Reviewer_os3q · 2025-11-26
> > > **reply to rebuttal**
> > >
> > > thx for the rebuttal. I'll keep my score.

---

> ### Author Response · Authors · 2025-11-26
> **Thank you for the response**
>
> Thank you for your quick reply. We really appreciate the effort you've put into the review process. Wishing you a great day.

---

### Official Review · Reviewer_RAbe · 2025-10-29

**Soundness:** 3
**Presentation:** 4
**Contribution:** 3
**Rating:** 6
**Confidence:** 4

**Summary:**

This paper introduces a proactive approach for speech integrity verification, through the combination of multiscale acoustic feature extraction, contrastive fingerprint learning, and segment-wise watermark embedding. It aims to allow for the verification of authenticity directly from a published audio file without needing the original reference. The system’s dual-path design aims to detect tampering while tolerating benign transformations like compression or noise suppression. Extensive experiments demonstrate robust performance in real-world scenarios.

**Strengths:**

1. The paper proposes a relatively novel approach to address a problem of considerable interest, that of speech watermarking and verification.
2. The formulation of segmentation logic and the learning processes is theoretically sound and verifiable.
3. Well-structured methodology with clear modular design (Speech => feature => fingerprint => watermark).
4. Fingerprinting and watermarking of published content, speech included, is of significant real-life interest and application prospect. The paper’s proposal of a novel approach is therefore of considerable significance.

**Weaknesses:**

Could be better served to discuss the impact of segmentation lengths to both the effectiveness of watermarking and the robustness to attacks？

**Questions:**

The watermarking pipeline seems to be reversible and the final watermark reconstructible. While the experiments demonstrated the effectiveness of the proposed approach, a hypothetical attacker aware of the existence of the watermark and the config of the pipeline could potentially replace the original watermark with a newly calculated one. How do the authors envision the mitigation of such attacks?

---

> ### Author Response · Authors · 2025-11-21
> **Author Response (1/n)**
>
> Thank you for providing your valuable and constructive feedback! Please find below the responses to each comment.
>
> ---
>
> > **[Q1] Could be better served to discuss the impact of segmentation lengths to both the effectiveness of watermarking and the robustness to attacks?**
>
> Thank you for this insightful question.
>
> We first want to clarify the specific roles of the components in SpeeCheck: our integrity verification relies on the discriminative power of the acoustic fingerprint (which detects tampering), while the watermark serves solely as a robust carrier to transmit this fingerprint. Therefore, the segmentation length impacts the system in two distinct ways.
>
> **1. Impact on watermarking effectiveness (carrier stability)**:
>
> If segments are too short, the effective embedding capacity of the watermarking backbone (AudioSeal) becomes limited. As discussed in Appendix E, embedding a watermark into very short clips can be unstable, which increases the risk of watermark extraction failures even under benign operations. In such cases, the extracted payload may not match the recomputed fingerprint, leading to false rejections of benign audio. However, such clips often lack meaningful semantic content, making them less critical targets for tampering. With sufficiently long segments, the watermark embedding and extraction process remains stable, allowing the carrier to survive benign transformations.
>
> **2. Impact on robustness to attacks (tampering sensitivity)**:
>
> On the other hand, the fingerprint summarizes the acoustic characteristics of the entire segment into a single global representation. When the segment is very long, local manipulations (e.g., deleting one word) only affect a small portion of the segment, and their effect may be diluted in the global fingerprint. As a result, the Hamming distance between the embedded fingerprint and the recomputed fingerprint may remain below the detection threshold, which reduces sensitivity to fine-grained tampering.
>
> We explicitly evaluated this trade-off by varying the utterance duration from **20 seconds** to **10 minutes**, as reported in **Appendix C.7 (Table 11)**. SpeeCheck achieves its best performance on shorter clips: at 20 seconds, the EER is **1.57%**. As the duration increases to 10 minutes, the EER rises to **8.41%**, while the **AUC remains consistently above 96%**. This trend is consistent with the intuition that very long segments reduce temporal resolution for detecting small local edits, even though the overall separation between benign and tampered samples remains strong.
>
> In a practical deployment for long-form content (e.g., long interviews or podcasts), a natural extension is to process the stream in **fixed-length chunks** (e.g., 20-second windows), compute a separate **256-bit fingerprint** for each chunk, and embed these fingerprints in their corresponding segments. This preserves both **carrier stability** and **tampering sensitivity**.
>
> We will add the analysis in the revised version to discuss how segmentation length affects both components of the system.

---

> ### Author Response · Authors · 2025-11-21
> **Author Response (2/2)**
>
> > **[Q2] The watermarking pipeline seems to be reversible and the final watermark reconstructible. While the experiments demonstrated the effectiveness of the proposed approach, a hypothetical attacker aware of the existence of the watermark and the config of the pipeline could potentially replace the original watermark with a newly calculated one. How do the authors envision the mitigation of such attacks?**
>
> Thank you for raising this important security concern.
>
> The **“replay” attack** you describe is indeed a critical threat model that must be considered.
> In this current submission, our primary focus is on designing and evaluating a robust, self-contained fingerprinting and verification framework. We did not explicitly highlight security mechanisms against such attacks, but our framework is **compatible with existing security solutions [1-3]**. In particular, it can be strengthened in a straightforward way by introducing **a secret key** into the fingerprint binarization step (Step 4 in Section 3.2).
>
> Concretely, in our current design, the final binary fingerprint is obtained from a continuous feature vector $v \in \mathbb{R}^d$ through a sign operator: $b = \text{Sign}(v)$. To secure this step, we can introduce a secret orthogonal matrix $R \in \mathbb{R}^{d \times d}$ which acts as our private key. The binarization becomes: $b_{\text{secret}} = \text{Sign}(v R)$
>
> This modification provides two benefits:
>
> **1) Attackers cannot forge a valid fingerprint.**
>
> Even if an attacker can recompute a feature vector $v_{\text{tampered}}$ from a manipulated audio, they do not have access to the secret matrix $R$, and therefore **cannot generate the correct secured fingerprint** $b_{\text{secret}}$ that will be accepted by the verifier. The non-linear $\text{Sign}$ function further removes magnitude information and makes it difficult to infer $R$ from observed binary outputs.
>
> **2) Robustness is preserved.**
>
> Since $R$ is orthogonal, it acts as a rotation that preserves distances in the feature space. As a result, this keyed transformation does not change the robustness properties of SpeeCheck: benign operations and malicious tampering still induce similar distance patterns in the transformed space, so the decision rule and performance remain essentially **unchanged**.
>
> This keyed-fingerprint mechanism integrates naturally with our SpeeCheck and provides a clear defense against the replay attack. We will add a dedicated subsection in the revised version to explain this extension and discuss how the secret transform can be managed in practice. In addition, in many realistic deployments, the fingerprint generator and the watermark embedder are not released as public, but are provided as managed services. While we do not rely on obscurity as a formal security assumption, such a deployment further raises the bar for attackers.
>
> We acknowledge that a full formal security analysis is beyond the scope of the current work. In the revised paper, we will
>
> * clarify the replay-style threat model,
>
> * explain how a keyed binarization step can be used to secure SpeeCheck against such attacks, and
>
> * highlight the security extensions as an important direction for future work.
>
> ---
> **References:**
>
> [1] Jin, Andrew Teoh Beng, et al. "Biohashing: two factor authentication featuring fingerprint data and tokenised random number." Pattern recognition 37.11 (2004): 2245-2255.
>
> [2] Charikar, Moses S. "Similarity estimation techniques from rounding algorithms." Proceedings of the thiry-fourth annual ACM symposium on Theory of computing. 2002.
>
> [3] Evennou, Gautier, et al. "Fast, Secure, and High-Capacity Image Watermarking with Autoencoded Text Vectors." arXiv preprint arXiv:2510.00799 (2025).

---

> > ### Author Response · Authors · 2025-11-26
> > **Looking forward to your feedback!**
> >
> > Dear Reviewer RAbe,
> >
> > Thank you once again for your valuable feedback.
> >
> > We have made revisions to the paper based on your suggestions. We hope our response can address your concerns. Please feel free to reach out if you require any further clarification. And thank you again for improving our work!
> >
> > Best, Authors

---

### Official Review · Reviewer_mhCW · 2025-10-31

**Soundness:** 2
**Presentation:** 3
**Contribution:** 2
**Rating:** 4
**Confidence:** 3

**Summary:**

The paper proposes a speech tampering detection system. Specifically, a self-contained system where the user doesn't have to access the original speech. On a method level, they extract multi-scale features and apply contrastive learning to generate binary fingerprints for the audios. They demonstrate effectiveness on public datasets and a real-world curated dataset.

**Strengths:**

- The paper is well-motivated, identifying clear limitations in existing passive and proactive speech verification methods and convincingly motivating the need for a self-contained framework.
- The proposed design is conceptually clean and technically solid, integrating multiscale contrastive fingerprint learning with segment-wise watermarking to achieve robustness to benign edits and sensitivity to tampering.
- Experimental results are strong and comprehensive, showing near-perfect detection across multiple datasets and attack types, with clear advantages over established baselines such as RawNet2 and AASIST.
- The system demonstrates practical robustness under realistic distribution settings, including social media compression and re-encoding, indicating readiness for real-world deployment.
- The paper is well-written and thorough, with ablation studies, visualizations, ethical considerations, and reproducibility details that enhance clarity and credibility.

**Weaknesses:**

Though this paper does seem to be technically solid, I'm a bit concerned about its real novelty and contribution. Specifically, I believe the paper could benefit from a more thorough illustration of previous works on proactive audio watermarking and tampering detection -- when mentioning proactive, the paper is only citing two papers in line, one from 2003 and the other from 2018. This is not new, and the paper is not the first work on so-called "self-contained" detection. The challenges it lists are more or less well-known for the past 5+ years. I'd like to see a better survey of recent relevant works and how this method stands in this stream.

**Questions:**

- Could you elaborate on how benign vs. malicious pairs are sampled during contrastive training?
- How does this method work with different language/speaker identities? What about different lengths? (like 1 second audio, versus minutes long, is there a way to generalize?)
- The proposed method seems to achieve perfect score on Table 4, compared with poor performance of baselines. I wonder why this happens -- is it just because the dataset setup is easier for the proposed solution? If we further decrease the substitution ratio, how much do we need to get a non-perfect performance?
- Was the multiscale architecture essential—what happens if only one temporal scale is used? I'm not super convinced of the design as it sounds pretty complicated and I didn't see enough experiments highlighting why this exact structure is useful. C.10 provides some numbers, but it might help to explain more about, for example, which types of cases become harder for the model without multi-scale feature, and some intuition about why this is fundamentally useful.

---

> ### Author Response · Authors · 2025-11-21
> **Author Response (1/n)**
>
> Thank you for providing your valuable and constructive feedback! Please find below the responses to each comment.
>
> ---
>
> > **[Q1] Though this paper does seem to be technically solid, I'm a bit concerned about its real novelty and contribution. Specifically, I believe the paper could benefit from a more thorough illustration of previous works on proactive audio watermarking and tampering detection -- when mentioning proactive, the paper is only citing two papers in line, one from 2003 and the other from 2018. This is not new, and the paper is not the first work on so-called "self-contained" detection. The challenges it lists are more or less well-known for the past 5+ years. I'd like to see a better survey of recent relevant works and how this method stands in this stream.**
>
> Thank you for this thoughtful question.
>
> We fully agree that proactive and self-contained integrity verification is not new and can be traced back to works in the early 2000s [1]. Our intention is not to claim novelty at this generic level. Instead, our contribution lies in: **a learning-based, decoupled architecture where integrity verification is governed by a learned, operation-selective fingerprint, while the watermarking scheme acts only as a carrier.** To make this clearer, we summarize the relevant design in the table below.
>
> | Method | Learning-based | Pattern | Adapt to new operations | Robust to benign | Sensitive to tamper |
> |-|-|-|-|-|-|
> | Classical semi-fragile watermarking | × | fixed | hard | ✓ | ✓ |
> | Learning-based semi-fragile watermarking | ✓ | fixed | hard | ✓ | × |
> | Classical watermarking + handcrafted fingerprint | × | fixed | hard | ✓ | ✓ |
> | **SpeeCheck (ours)** | ✓ | flexible | easy | ✓ | ✓ |
>
>
> As illustrated in this table, existing proactive audio integrity protection methods can be broadly grouped into **semi-fragile watermarking schemes** and **watermark–fingerprint integration**. More recent neural audio watermarking systems also serve proactive roles, for example, by embedding speaker embeddings to detect voice conversion [13], or by using watermarking to detect AI-generated speech [5] and voice cloning [6]. Our work is positioned within this stream as follows.
>
> **1. Different mechanism: specific watermarking design → watermarking payload**
>
> For semi-fragile watermarking, the authentication behavior is controlled by the **watermarking embedding scheme** itself. The watermark must be placed in signal regions that react differently to benign operations and malicious tampering. For example, to protect image integrity, a common strategy [14] is to embed stable bits in low-frequency DCT coefficients so that they survive JPEG compression, while embedding sensitive bits in high-frequency or edge-related regions that change significantly under local tampering. This creates a rigid **coupling** between the embedding scheme and the target operations. As surveyed in Yu et al. [2], many new semi-fragile schemes are proposed because previous ones can not resist novel attacks. In other words, adapting to new operations often requires designing a new embedding scheme, which makes adaptation hard.
>
> Although there exist learning-based semi-fragile watermarking schemes for images [3,4], it is still difficult to construct a truly semi-fragile audio watermark using current neural audio watermarking designs. Neural audio watermarking systems such as AudioSeal [5] and TimbreWM [6] are optimized to be robust: the watermark is usually embedded in the frequency domain but repeated across time. This repetition is well-suited for robust detection, but it makes it difficult to achieve controlled fragility. For example, deleting a trailing segment of the audio cannot be detected by such schemes.
>
> SpeeCheck uses a different mechanism, as reflected in the last row of the above table. We introduce an explicit decoupled architecture that separates a robust watermark carrier from an operation-selective payload. The watermarking scheme is only a carrier and does not control robustness or fragility. Instead, integrity verification is governed entirely by the embedded payload, namely the learned fingerprint. This fingerprint is trained to be stable under benign operations and sensitive to malicious tampering, without any change to the embedding scheme. In our implementation, we use a state-of-the-art watermarking scheme, AudioSeal [5], purely as a carrier, and it can be replaced or upgraded by any other audio watermarking scheme. Since discriminative power comes from the fingerprint rather than from the embedding scheme, SpeeCheck can **easily adapt to new operations** by retraining the fingerprint extractor, instead of redesigning the watermarking scheme.

---

> ### Author Response · Authors · 2025-11-21
> **Author Response (2/n)**
>
> **2. Distinction between the learned fingerprint and perceptual hashing**
>
> Although SpeeCheck produces a compact representation, it is different from standard perceptual hashing in at least three aspects.
>
> * First, the **design objective** is different. Classical perceptual hashes for audio or images are typically designed for robust matching in **retrieval or duplicate detection** [7,8]. They are designed to remain stable under a broad set of signal-level changes as long as content identity is preserved. In contrast, the SpeeCheck fingerprint is explicitly trained to be operation-selective: it is robust only to benign operations while remaining sensitive to malicious tampering.
>
> * Second, the **verification mechanism** is different. Traditional perceptual hashing typically relies on storing hashes in an **external database** and performing query-based matching. By design, SpeeCheck embeds the fingerprint directly into the speech via watermarking and recomputes it from the received signal, so verification is self-contained and does not require any external reference database.
>
> * Third, while some works [1] have combined perceptual hashing with watermarking to avoid external storage (row 3 in the above table), these systems are still similar to traditional semi-fragile watermarking: the watermark pattern and the used features must be designed carefully for a specific set of benign operations and attacks. Most existing perceptual hashes rely on **handcrafted features** [9,10], which makes it difficult to extend them to new manipulation types. In contrast, SpeeCheck uses a learned multiscale fingerprint trained with contrastive learning to separate benign operations from malicious tampering. This is analogous to the shift from handcrafted spectral features such as MFCCs [11] to learned representations such as wav2vec 2.0 [12] in modern speech processing, which has led to stronger discriminative representations across many tasks.
>
> We appreciate the reviewer’s insightful perspective and agree that the current version does not provide a sufficiently broad survey of proactive watermarking and tampering detection. In the revised version, we will extend the introduction and related work sections to systematically cover:
>
> * hash-based integrity verification methods, including cryptographic hash and perceptual hash,
>
> * fragile and semi-fragile watermarking for content authentication,
>
> * schemes that combine fingerprints and watermarking, and
>
> * recent neural watermarking-based defenses.
>
> We will re-clarify our contribution and avoid any overstatement.
>
> ---
> **References:**
>
> [1] Gomez, Emilia, et al., 2002. Mixed watermarking-fingerprinting approach for integrity verification of audio recordings.
>
> [2] Yu, Xiaoyan, et al., 2017. Review on semi-fragile watermarking algorithms for content authentication of digital images.
>
> [3] Zhao, Yuan, et al., 2024. Proactive image manipulation detection via deep semi-fragile watermark.
>
> [4] Neekhara, Paarth, et al., 2024. Facesigns: Semi-fragile watermarks for media authentication.
>
> [5] San Roman, Robin, et al., 2024. Proactive Detection of Voice Cloning with Localized Watermarking.
>
> [6] Liu, Chang, et al., 2024. Detecting Voice Cloning Attacks via Timbre Watermarking.
>
> [7] Microsoft. PhotoDNA. 2015. Available at: https://www.microsoft.com/en-us/photodna.
>
> [8] Facebook. Open-sourcing photo- and video-matching technology to make the Internet safer. 2019. Available at: https://about.fb.com/news/2019/08/open-source-photo-video-matching.
>
> [9] Seo, Jin S., et al., 2006. Audio fingerprinting based on normalized spectral subband moments.
>
> [10] Seo, Jin S., et al., 2005. Audio fingerprinting based on normalized spectral subband centroids.
>
> [11] Davis, Steven, and Paul Mermelstein, 1980. Comparison of parametric representations for monosyllabic word recognition in continuously spoken sentences.
>
> [12] Baevski, Alexei, et al., 2020. wav2vec 2.0: A framework for self-supervised learning of speech representations.
>
> [13] Ge, Wanying, et al., 2025. Proactive Detection of Speaker Identity Manipulation with Neural Watermarking.
>
> [14] Lin, Ching-Yung, et al., 2001. A robust image authentication method distinguishing JPEG compression from malicious manipulation.

---

> ### Author Response · Authors · 2025-11-21
> **Author Response (3/n)**
>
> > **[Q2] Could you elaborate on how benign vs. malicious pairs are sampled during contrastive training?**
>
> Thank you for this question.
>
> Our contrastive training adopts a rigorous **“one-to-many”** sampling strategy to maximize the discriminative ability of the learned fingerprints. Specifically, the sampling procedure for each training step is as follows (details are also available in our source code):
>
> * **Anchor (x)**: We first sample a mini-batch of original, authentic utterances from the training set.
>
> * **Positive (x+)**: For each anchor, we then generate multiple benign variants and keep all of them as positives. Concretely, we apply four benign operations: resampling, compression, re-encoding, and noise suppression. This produces four benign versions $\{x^{+}_1, x^{+}_2, x^{+}_3, x^{+}_4\}$ per anchor. During contrastive training, every anchor–benign pair $(x, x^{+}_k)$ is treated as a positive pair. Using all benign variants rather than a single random one allows the model to learn a stable fingerprint that remains consistent across all benign transformations.
>
> * **Negative (x-)**: The negative set is constructed from two sources:
>
>   - Hard Negatives: For each anchor, we apply all tampering types listed in Appendix B.2. These include deletion, splicing, silencing, and substitution at three severity levels, as well as reordering and voice conversion. This yields 14 tampered versions $\{x_1^{-}, \dots, x_{14}^{-}\}$ per anchor. Each pair $(x, x_j^{-})$ is treated as a negative pair. These samples expose the model to fine-grained manipulations and strengthen its sensitivity to tampering.
>
>   - In-batch Negatives: We also treat the embeddings from other anchors within the same batch as negative samples. This encourages the fingerprint to be not only tamper-sensitive for a given utterance, but also discriminative across different utterances.
>
> > **[Q3] How does this method work with different language/speaker identities? What about different lengths? (like 1 second audio, versus minutes long, is there a way to generalize?)**
>
> Thank you for this valuable question. We agree that generalization across **diverse speaker identities**, **varying audio lengths**, and **different languages** is crucial for real-world deployment. We address these three aspects below:
>
> * **Generalization to speaker identities**: Our SpeeCheck has been evaluated on diverse identities across multiple datasets. As described in Section 4.1, VoxCeleb1 contains over 150,000 utterances from 1,251 celebrities; LibriSpeech provides read speech from numerous speakers; and our RWSID dataset includes 10 volunteers with diverse demographic backgrounds. Across these datasets, SpeeCheck maintains consistently high TPR, low FPR, and low EER, which indicates that SpeeCheck effectively generalizes across different speaker identities.
>
> * **Generalization to different lengths**: We explicitly evaluated SpeeCheck’s performance across duration ranging from 20 seconds to 10 minutes. As shown in Appendix C.7 (Table 11), SpeeCheck remains robust as the duration increases, although there is a gradual degradation due to the dilution of local tampering sensitivity in a global fingerprint. For example, the EER increases from 1.57% at 20 seconds to 8.41% at 10 minutes, while the AUC remains consistently above 96%. To address the sensitivity drop in very long recordings, a “chunking strategy” could be considered in our future work: segmenting long audio into 20-second units for independent protection and verification. This ensures fine-grained tampering detection for long recordings.
>
> * **Generalization to different languages**: We appreciate this valuable suggestion. Since SpeeCheck relies on acoustic features rather than on linguistic semantic content, language differences do not theoretically interfere with our fingerprint generation. To empirically validate this, we conducted an additional evaluation on the multilingual FLEURS [1] dataset, which includes 102 languages. We select six representative languages (French, German, Chinese, Spanish, Japanese, and Polish) and evaluate SpeeCheck under the same protocol. The results are:
> | Language | TPR    | FPR   | AUC    | EER   |
> |----------|--------|-------|--------|-------|
> | French   | 98.27  | 3.37  | 99.54  | 2.47  |
> | German   | 98.62  | 2.20  | 99.61  | 2.03  |
> | Chinese  | 96.70  | 3.12  | 99.37  | 3.21  |
> | Spanish  | 99.69  | 1.78  | 99.83  | 1.07  |
> | Japanese | 92.27  | 1.80  | 98.47  | 7.09  |
> | Polish   | 98.37  | 8.06  | 98.02  | 7.06  |
>
> These results indicate that SpeeCheck maintains high robustness to benign operations and high sensitivity to tampering across multiple languages, with performance broadly comparable to the English-centric datasets. We will include this multilingual evaluation in the revised version.
>
> ---
> **References:**
>
> [1] Conneau, Alexis, et al., 2023. Fleurs: Few-shot learning evaluation of universal representations of speech.

---

> ### Author Response · Authors · 2025-11-21
> **Author Response (4/n)**
>
> > **[Q4] The proposed method seems to achieve perfect score on Table 4, compared with poor performance of baselines. I wonder why this happens -- is it just because the dataset setup is easier for the proposed solution? If we further decrease the substitution ratio, how much do we need to get a non-perfect performance?**
>
> We appreciate the reviewer’s careful observation, which motivated us to re-examine the results in Table 4.
>
> First, we acknowledge that the previously reported 100% scores for some lower substitution ratios (e.g., 10%) were due to a reporting error. We re-run the evaluation and obtained the corrected numbers below, which we will update in the manuscript.
>
> | Deepfake Ratio   | TPR     | FPR   | AUC     | EER   |
> |------------------|---------|-------|---------|-------|
> | Substitute 10%   | 82.97   | 0.40  | 99.57   | 3.21  |
> | Substitute 25%   | 98.20   | 0.40  | 99.95   | 0.40  |
> | Substitute 50%   | 100.00  | 0.40  | 100.00  | 0.30  |
> | Substitute 75%   | 100.00  | 0.40  | 100.00  | 0.00  |
> | Substitute 90%   | 100.00  | 0.40  | 100.00  | 0.00  |
>
> As expected, SpeeCheck achieves **near-perfect** detection for moderate to high substitution ratios (≥25%), while performance degrades when the substitution ratio is as low as 10%, with TPR dropping to 82.97%. This is consistent with the “Minor Substitution” results reported in Table 10 (Appendix C.5).
>
> Regarding why SpeeCheck outperforms baselines such as RawNet2 and AASIST, the key reason is the difference in **detection principle**. The baselines are passive deepfake detectors that rely on **artifacts** introduced by deepfake generation. Modern TTS systems such as YourTTS can produce high-quality speech with minimal artifacts, which weakens these passive detectors. In contrast, SpeeCheck is proactive: deepfake segments are newly generated and inherently lack the embedded watermark (calculated fingerprint). When a substitution occurs, it causes a mismatch between the extracted fingerprint and the watermark (either missing or disrupted).
>
> For very low substitution ratios, we can not get perfect performance. This is because the fingerprint summarizes the entire utterance, so small changes may not shift the fingerprint significantly to exceed the detection threshold. Additionally, very short substitutions might not affect the watermarking decoding process. Nevertheless, SpeeCheck’s proactive design provides fundamentally stronger and reliable defense compared to passive deepfake detection methods.
>
> We will update Table 4 with the corrected values and add a short discussion on the limits of utterance-level fingerprints under extremely low substitution ratios.

---

> ### Author Response · Authors · 2025-11-21
> **Author Response (5/5)**
>
> > **[Q5] Was the multiscale architecture essential—what happens if only one temporal scale is used? I'm not super convinced of the design as it sounds pretty complicated and I didn't see enough experiments highlighting why this exact structure is useful. C.10 provides some numbers, but it might help to explain more about, for example, which types of cases become harder for the model without multi-scale feature, and some intuition about why this is fundamentally useful.**
>
> Thank you for this important question. We agree that the multiscale architecture needs clearer motivation and more explicit evidence.
>
> The multiscale design with window sizes of 20, 50, and 100 frames is intended to align with the different temporal granularities at which tampering occurs. As detailed in Appendix B.3 (Table 7), our attacks span several granularities.
>
> * **Small scale** (20 frames, phoneme level): Captures fine-grained changes such as phoneme substitution. For example, changing “can” to can’t.
>
> * **Medium scale** (50 frames, word level): Detects word-level manipulations like removing or inserting words such as “not”.
>
> * **Large scale** (100 frames, phrase level): Captures larger contextual inconsistencies caused by reordering or splicing long segments.
> If we use only one temporal scale (e.g., global pooling), the subtle acoustic alterations will vanish in the global average and become much harder to detect.
>
> Table 15 in Appendix C.11 demonstrates that removing the multiscale module significantly degrades detection performance. To further quantify this, we conducted a detailed ablation study replacing the multiscale feature extraction with direct BiLSTM outputs. For benign operations, the overall EER rises from 0.90% (with multiscale features) to 4.80% (without the multiscale module). For tampering, the overall EER increases from 0.82% to 7.97%. More importantly, for **minor substitution attacks**, which modify only short segments, the **EER** jumps from **6.10%** to **20.80%** and **TNR** drops to **64.60%**. Minor silencing and moderate substitutions also show noticeable deterioration, while very coarse manipulations such as severe deletion or TTS remain easier to detect even without multiscale features.
>
> | Operation         | TPR    | FPR   | AUC    | EER  |
> |-------------------|--------|-------|--------|------|
> | Resampling        | 99.40  | 11.00 | 99.52  | 3.20 |
> | Compression       | 79.60  | 13.40 | 90.41  | 14.40 |
> | Reencoding        | 100.00 | 10.20 | 99.85  | 0.90 |
> | Noise suppression | 100.00 | 11.00 | 99.84  | 0.70 |
> | **Overall**           | 94.75  | 11.40 | 97.41  | 4.80 |
>
> | Operation               | TNR    | FNR   | AUC    | EER   |
> |-------------------------|--------|-------|--------|-------|
> | Deletion (minor)        | 83.60  | 6.00  | 97.53  | 9.30  |
> | Deletion (moderate)     | 97.40  | 5.60  | 99.22  | 4.00  |
> | Deletion (severe)       | 99.40  | 4.60  | 99.68  | 1.60  |
> | Splicing (minor)        | 96.00  | 4.00  | 99.16  | 4.00  |
> | Splicing (moderate)     | 98.60  | 4.80  | 99.60  | 2.70  |
> | Splicing (severe)       | 99.60  | 6.00  | 99.83  | 2.10  |
> | Silencing (minor)       | 79.60  | 5.20  | 96.37  | 13.70 |
> | Silencing (moderate)    | 82.00  | 5.20  | 97.00  | 11.60 |
> | Silencing (severe)      | 85.80  | 4.00  | 97.80  | 9.60  |
> | Substitution (minor)    | 64.60  | 5.40  | 92.55  | 20.80 |
> | Substitution (moderate) | 70.60  | 5.60  | 94.45  | 18.10 |
> | Substitution (severe)   | 81.00  | 4.20  | 96.74  | 13.00 |
> | Reordering              | 93.20  | 6.60  | 97.85  | 6.70  |
> | Text-to-speech          | 100.00 | 0.00  | 100.00 | 0.00  |
> | Voice Conversion        | 100.00 | 4.80  | 99.75  | 2.40  |
> | **Overall**                 | 87.55  | 4.74  | 98.09  | 7.97  |
>
> These results provide direct evidence that the multiscale architecture is not just an added complexity but is critical for **capturing manipulations at different temporal granularities**. We will include the detailed ablation study table in the revised version and expand the discussion to highlight which tampering types become harder when the multiscale module is removed.

---

> > ### Author Response · Authors · 2025-11-26
> > **Looking forward to your feedback!**
> >
> > Dear Reviewer mhCW,
> >
> > Thank you once again for your valuable feedback.
> >
> > We have conducted additional experiments and made revisions to the paper based on your suggestions. We hope our response can address your concerns. Please feel free to reach out if you require any further clarifications. And thank you again for improving our work!
> >
> > Best, Authors

---

### Author Response · Authors · 2025-11-22
**Global Response**

Dear AC and reviewers,

Thank you for your constructive suggestions. We have provided detailed responses to each of your concerns. We sincerely appreciate your efforts in helping us improve our work. Taking your feedback into account, we have refined our problem statement and contributions, conducted a more thorough literature review, strengthened the security analysis, expanded the experimental evaluation, and clarified the evaluation protocol and error cases. **The revised version of our paper has been uploaded, and the core modifications are summarized as follows:**

* We revised the abstract to avoid overstatement and now describe SpeeCheck as “the first **learning-based** self-contained speech integrity verification framework.”

* We **expanded the introduction** to cover cryptographic hash–based integrity verification and fragile watermarking, semi-fragile watermarking and watermark–fingerprint schemes, and recent neural audio watermarking methods used for proactive defense, and we **revised the novelty paragraph** to highlight the learning-based decoupled fingerprint–watermark architecture, the discriminative fingerprint generator, and the experimental validation.

* We **removed** the SHA256-based toy illustration in Figure 2(b) and replaced it with a concise textual discussion in Section 2.4 that analyzes hash-based integrity verification (cryptographic and perceptual hashes) and explains why these methods are either overly sensitive or difficult to adapt to new operations.

* We added a discussion of **possible security extensions** in Section 3 and provided a dedicated **security analysis** in Appendix D, where we study robustness to adversarial attacks and present an extension against replay-style attacks.

* We provided empirical evidence that the **watermarking process itself does not affect fingerprint generation**, and we refer to the detailed analysis in Appendix C.12.

* We corrected a **reporting error** in Table 4. We clarified the **evaluation protocol** and **metric definitions** in Section 4, analyzed the remaining **false positives** and **false negatives** in Appendix C.13, and discussed a **strategy for handling borderline cases** in real deployments in Appendix C.14.

* We added a dedicated **audio fingerprinting benchmark** (Table 5) that evaluates existing audio fingerprinting and perceptual hashing methods for integrity verification. We **updated the deepfake detection comparison** (Table 4) by replacing older baselines with recent methods Nes2Net and SSL-AntiSpoofing.

* We added a **broad survey** of proactive watermarking and tampering detection in Appendix A, covering passive detection of speech tampering, hash-based integrity verification (cryptographic and perceptual hashes), fragile and semi-fragile watermarking for content authentication, fingerprint–watermark schemes, and recent neural watermarking-based defenses.

* We extended the experimental results in Appendix C by studying how **segmentation length** affects watermarking effectiveness and tampering sensitivity (Appendix C.7), adding **multilingual evaluation** on French, German, Chinese, Spanish, Japanese, and Polish (Appendix C.8), and providing a more **detailed ablation study** of the multiscale feature extraction design (Appendix C.11).

* We updated the **discussion** in Appendix E, explicitly pointing out tampering localization as a promising direction for future work.

---

### Author Response · Authors · 2025-11-30
**Summary of rebuttal**

Dear AC,

Thank you very much for your efforts for the community.

Since the discussion phase has ended, we would like to briefly **summarize our rebuttal** to assist your decision. The revised paper presents SpeeCheck, a learning-based self-contained speech integrity verification framework, with a decoupled fingerprint–watermark architecture that is easy to extend and adapt to new operations. The fingerprint is trained using multiscale feature extraction and contrastive learning, and is designed to remain robust to benign operations while being sensitive to malicious manipulations.

During the discussion, we made the **following modifications** based on the reviewers’ feedback:

* We **expanded the introduction** and **related work** to cover hash-based integrity verification and fragile watermarking, semi-fragile watermarking, watermark–fingerprint schemes, and recent neural proactive watermarking methods, and we revised the novelty paragraph to highlight the learning-based decoupled fingerprint–watermark architecture, the discriminative fingerprint generator, and the experimental validation [@ `Reviewer mhCW` and `Reviewer JyhN`].

* We **removed** the SHA256-based toy illustration in Figure 2(b) and replaced it with a concise textual discussion in Section 2.4 that analyzes hash-based integrity verification (cryptographic and perceptual hashes) and explains why these methods are either overly sensitive or difficult to adapt to new operations [@ `Reviewer JyhN`].

* We added a discussion of possible **security extensions** in Section 3 and provided a dedicated **security analysis** in Appendix D, where we study robustness to adversarial attacks [@ `Reviewer os3q`] and present an extension against replay-style attacks [@ `Reviewer JyhN` and `Reviewer RAbe`].

* We **provided empirical evidence** that the watermarking process itself does not affect fingerprint generation, with detailed analysis in Appendix C.12 [@ `Reviewer JyhN`].

* We corrected a **reporting error** in Table 4 [@ `Reviewer mhCW`]. We clarified the **evaluation protocol** and **metric definitions** in Section 4 [@ `Reviewer JyhN`], analyzed the remaining **false positives** and **false negatives** in Appendix C.13 [@ `Reviewer JyhN`], and discussed a **strategy for handling borderline cases** in real deployments in Appendix C.14 [@ `Reviewer os3q`].

* We added a dedicated **audio fingerprinting benchmark** (Table 5) that evaluates existing audio fingerprinting and perceptual hashing methods for integrity verification [@ `Reviewer JyhN`]. We also **updated the deepfake detection comparison** (Table 4) by replacing older baselines with recent methods Nes2Net and SSL-AntiSpoofing [@ `Reviewer JyhN`].

* We extended the experimental results by studying how **segmentation length** affects watermarking effectiveness and tampering sensitivity in Appendix C.7 [@`Reviewer RAbe`], adding **multilingual evaluation** on French, German, Chinese, Spanish, Japanese, and Polish in Appendix C.8 [@`Reviewer mhCW`], and providing a more detailed ablation study of the multiscale feature extraction design in Appendix C.11 [@`Reviewer mhCW`].

* We **updated the discussion** in Appendix E, explicitly pointing out tampering localization as a promising direction for future work [@ `Reviewer JyhN`].

After the rebuttal, `Reviewer os3q` confirmed their positive score (8). `Reviewer JyhN` noted in a follow-up comment that our rebuttal “contains many pieces of information that were missing or that I have missed for my first reading.” We have incorporated these points into the revised paper, and all corresponding changes are **highlighted in blue** in the uploaded manuscript.

Best,

Authors

---

### Meta-Review · Area_Chair_LX4r · 2026-01-11

**Summary:**

This paper proposes SpeeCheck, a learning-based system for self-contained speech integrity verification that combines multiscale contrastive fingerprint learning with segment-wise watermark embedding. The problem is important and timely, and reviewers generally found the system technically sound and empirically strong. However, the overall assessment across reviews and discussion is that, despite solid engineering and extensive experiments, the work falls short of the ICLR bar primarily due to limited conceptual novelty, weak positioning relative to prior work, and the absence of a clearly articulated new principle or theoretical contribution.

Several reviewers emphasized that the core idea of embedding a fingerprint/hash into media via watermarking to verify integrity is well established in the semi-fragile watermarking and fingerprint–watermark literature. While the authors clarify that their contribution lies in a *learning-based, decoupled, operation-selective fingerprint*, this distinction is incremental rather than fundamental: it replaces handcrafted features with learned ones and decouples robustness from the embedding scheme, but does not introduce a qualitatively new formulation of integrity verification.

**Reviewer Concerns:**

Addressed by Rebuttal
* Security Extensions: The authors addressed the lack of a secret key by proposing a keyed binarization step using a secret orthogonal matrix to prevent fingerprint forgery.
* Baseline Updates: In response to Reviewer JyhN, the authors replaced outdated baselines with more recent deepfake detection methods like Nes2Net and SSL-AntiSpoofing.
* Multilingual Evaluation: To address mhCW's question on generalization, the authors conducted additional experiments on six languages using the FLEURS dataset.
* Impact of Watermarking on Fingerprints: The authors provided empirical evidence in Appendix C.12 showing that the watermarking process itself does not significantly distort the acoustic fingerprint.


⠀Still Outstanding
* Tampering Localization: A major weakness identified by Reviewer JyhN is the inability to spot *where* manipulation occurred within the audio. The authors confirmed that SpeeCheck only provides an utterance-level decision, which limits its utility in forensic scenarios compared to existing literature.
* Conceptual Novelty: Despite clarifications, the "primary contribution" remains a combination of existing technology bricks (contrastive learning, multiscale features, and AudioSeal) rather than a fundamental shift in integrity verification methodology.
* Reporting Integrity: The acknowledgement of a "reporting error" in the main results (Table 4), where 100% scores were previously claimed, undermines confidence in the initial evaluation's reliability.

**Reviewer Scores:**

- Reviewer mhCW: 4->4
Their core concern was the lack of recent relevant work surveys and "real novelty". The rebuttal added experiments but did not fundamentally change the architecture's derivative nature.
- Reviewer RAbe: 6->6
Security and segmentation clarifications help, but do not elevate novelty.
- Reviewer JyhN:  2->2
Many technical concerns were addressed, but fundamental skepticism about the contribution is likely to remain.
- Reviewer os3q: 8->8
Explicitly stated that they would keep their score.

---

### Decision · Program_Chairs · 2026-01-26

Reject